# Innovative deep matching algorithm for stock portfolio selection using deep stock profiles

**Ganggang Guo**[1]*, **Yulei Rao**[1], **Feida Zhu**[2], **Fang Xu**[3]*

**1** Business School, Central South University, Changsha, China, **2** School of Information Systems, Singapore Management University, Stamford Road, Singapore, Singapore, **3** Xiangtan Central Hospital, Clinical Practice Base of Central South University, Xiangtan, China

* ganggangguo66@gmail.com (GG); wodedianziyouxiang88@126.com (FX)

## Abstract

Construction of a reliable stock portfolio remains an open issue in quantitative investment. Multiple machine learning models have been trained for stock portfolio selection, but their practical applicability remains limited due to the challenges posed by the characteristic of a low signal-to-noise ratio (SNR), the nature of time-series data, and non-independent identical distribution in financial data. Here, we transformed the stock selection task into a matching problem between a group of stocks and a stock selection target. We proposed a novel representation algorithm of stock selection target and a novel deep matching algorithm (TS-Deep-LtM). Then we proposed a deep stock profiling method to extract the optimal feature combination and trained a deep matching model based on TS-Deep-LtM algorithm for stock portfolio selection. Especially, TS-Deep-LtM algorithm was obtained by setting statistical indicators to filter and integrate three deep text matching algorithms. This parallel framework design made it good at capturing signals from time-series data and adapting to non-independent identically distributed data. Finally, we applied the proposed model to stock selection and tested long-only portfolio strategies from 2010 to 2017. We demonstrated that the risk-adjusted returns obtained by our portfolio strategies outperformed those obtained by the CSI300 index and learning-to-rank approaches during the same period.

## Introduction

Stock portfolio selection is to seek an optimal allocation of assets among securities such that the investment yields the most profitable returns, while decentralizing the investment risk. The pioneering research work on portfolio selection theory is the mean-variance (M-V) model [1], by which if a group of stocks are not perfectly correlated, portfolio diversification can keep returns hold while minimizing the variance or, equivalently, risk, and unsystematic risk can be zero theoretically. Inspired by Markowitz's M-V model, researchers have been using quantitative approaches to enrich and improve modern portfolio selection theory and practice methods [2–5]. One of the most widely used is multi-factor stock selection strategy. Traditional multi-factor approach uses a structured multi-factor risk model and follows three steps, including factor discovery, factor screening, and stock selection modeling, to obtain

**Data Availability Statement:** All relevant data are within the paper and its Supporting information files.

**Funding:** This work was supported by the National Natural Science Foundation of China (No.

71910107001). The funders had no role in study design, data collection and analysis, decision to publish, or preparation of the manuscript.

**Competing interests:** The authors have declared that no competing interests exist.

quantitative strategies [6]. With the promotion of artificial intelligence technology applications, machine learning algorithms have began to be applied in factor discovery and stock selection modeling [7–11]. Examples including discovering new factors from social media data with text mining technologies [8] and constructing machine learning stock selection models after transforming the stock selection problems into the classification, regression, clustering, or rule-based optimization problems [9–13]. Most of these studies focus on building stock portfolios by stock selection models that target the predictions of absolute indicators, such as returns, trading volumes, and volatilities [14, 15].

However, these stock selection models have limited practical applicability due to the challenges posed by the characteristic of a low signal-to-noise ratio (SNR), the nature of time series, and non-independent identical distribution (Non-IID) in financial data [16]. Specifically, the prediction of stock returns can be regarded as a regression problem for time-series data. Neural network algorithms [17] and deep learning algorithms (e.g., RNNs and LSTMs) [18] are commonly used. The SNR is a good measure of distortion in time domain between original and reconstructed signal [19]. Financial data existing in a time-series format is usually accompanied by large fluctuations of non-stationary and unusual noise [20], resulting in its characteristic of a low SNR. Thus it is difficult for regression models to capture enough reconstructed signals in the training process so as to obtain final predictions with a high error rate. Alternatively, the prediction of the ups and downs of stock prices can be defined as a classification problem. Classification algorithms, such as support vector machines [21], random forests [22], and convolutional neural networks (CNNs) [23] are typically used. This kind of approach may achieve high accuracy, but it is not always accompanied by high returns from the trading strategy. For example, a correct prediction by the model may bring only a small gain; an occasional incorrect prediction may cause a great loss; thus, significant downside risks persist. Or, the prediction of the groups or clusters to which the stocks belong according to the price or returns can be considered as a clustering problem. Hierarchical clustering [24] and k-means clustering [25] are often used to construct stock selection models. This method can obtain a stock portfolio with similar volatility of returns, but the one with the highest predicted future returns can not be obtained directly. Lastly, the prediction of absolute indicators can be formulated as a rule-based optimization problem that derives the most effective quantization factors (e.g., indicators of techniques, fundamentals, and the macro economy) and the optimal algorithm parameters. Optimization algorithms include the genetic algorithm [26] and reinforcement learning [27]. However, due to model training relying on numerous parameters, empirical initialization, and long computing periods, this type of method is flawed. This method may lead to difficulty in training models using optimization algorithms and limited applicability. Faced with the deficiencies in implementing these models for quantitative investment, investigators have recently come to recognize that it is more feasible to use relative indicators as predictive targets of machine learning models. For instance, Song et al. [28] defined the stock selection task from the perspective of ranking investors' relative views on stocks' performance. The authors used learning-to-rank (LtR) [29], RankNet [30], and ListNet [31] to train ranking models. This method converted a prediction of stock returns into a comparison of returns between two stocks in a group of stocks, which enhanced the ability of models to capture reconstruction signals during the training process. In addition, the authors introduced the relative view of investors' sentiment as a novel predictive factor, which strengthened the reconstruction signals from training data, increased the SNR of training data, and thus improved the model's predictive performance [32]. Inspired by this finding, we used relative indicators as predictive targets of stock selection models and included multiple quantization factors including investors' sentiment into the feature combination of training data to mitigate the effects of low SNR in financial data to some extent. Nevertheless, the challenges posed by

the characteristics of time series and Non-IID in financial data persist. In reaction to these challenges, we transformed the stock selection task into a matching problem between a group of stocks and the stock selection target. We proposed a novel deep matching algorithm for stock portfolio selection using deep stock profiles.

The way to extract an optimal feature combination from the training data is directly related to the final performance of a machine learning model. Arbitrage pricing theory holds that stock returns are affected by multiple factors [3]. Researchers built on this theory by having revealed a series of multi-factor models that discovered the impact factors of cross-sectional stock returns. Out of these, the most representative one is the Fama-French three-factor model [5]. The quantitative factors discovered by these models have become a fundamental basis for extracting feature combinations from heterogeneous financial data [33, 34]. Empirical studies based on stock market and financial statement data in standard finance provided us with multiple traditional quantitative factors [35–37]. In digital information age, the social media represented by microblog, blog, network community, forum, etc. becomes the major platform of exchanging information and giving comments. The investors' mentality and perception of market information can directly affect their investment decisions, trading behaviors, and even financial markets [38–40]. This allows investigators to profile the impact on financial markets more accurately. New impact factors from social media data are constantly being discovered in empirical studies in behavioral finance [41, 42]. In addition to investor sentiment [43, 44], which has been widely demonstrated by behavioral finance researchers, investor attention [45, 46], investor disagreement [47], and social interactions [48] have been found to affect stock pricing. For example, Andrei et al. [47] pointed out that incomplete investor attention to market information in the current climate of information explosion could cause deviations in asset value prediction, being particularly common in China's securities market that was in transition and was dominated by retail investors. Hillert et al. [49] found that investor disagreement indicators were significantly negatively correlated with market returns the next day, which was especially more relevant during the recession. Additionally, the quantitative factors in behavioral finance from social media data have begun to be used in constructing trading strategies [50–52]. Nevertheless, only one single quantitative factor (e.g., investor sentiment) has been commonly used to generate multiple indicators and to design trading strategies in most studies. Manifold quantitative factors affecting cross-sectional stock returns have hardly been considered. In this study, we proposed a method of deep stock profiling for the first time, which could extract the optimal feature combination from heterogeneous financial data. The stock profiles consisted of both traditional and new quantitative factors, which could systematically represent the risk characteristics of stocks.

In addressing our defined matching problem, we innovatively proposed representation algorithms of stock features and stock selection targets, devised a deep matching algorithm, and trained a deep matching model for stock portfolio selection. Specifically, based on stock profiles, the feature vector of each stock was expressed as $X_{1 \times m}$, where $m$ represented the dimensions of vectors from stock profiles. Considering that financial data occurs in a time-series format, we went back by $(w - 1)$ windows in the historical feature vectors of each stock, and the feature representation of each stock was extended as stock feature matrix $X_{w \times m}$. This form of stock feature representation enabled deep learning algorithms to capture as many effective signals as possible during the training process. Then we designed a representation algorithm according to investment returns to generate the stock selection target matrices. By analogy with a question-answering problem [53], a stock feature matrix or a stock selection target matrix is equivalent to the representation of each answer or question, respectively. Therefore, deep text matching algorithms [54] could be introduced to address our matching problem. An innovative deep matching algorithm (deep learning-to-match for time series,

TS-Deep-LtM) was devised to train the stock matching model. The TS-Deep-LtM algorithm was obtained by setting statistical indicators to filter and integrate three deep text matching algorithms adapted for different data distribution characteristics. The parallel framework design of TS-Deep-LtM made it better at capturing signals from time-series data and adapting to Non-IID data. Noticeably, in model training, the strategy of dividing training data by samples enabled models to be fully learned on limited training data. Finally, we applied the TS-Deep-LtM algorithm to stock selection and tested long-only portfolio strategies using eight years of stock market, financial statement, and financial social media data. Through backtesting of these strategies from 2010 to 2017, we demonstrated that the risk-adjusted returns obtained by our portfolio strategies outperformed those obtained by both the CSI300 index and classical LtR approaches during the same period. The primary contributions of this study are to efficiently alleviate the challenges posed by the characteristics of a low SNR, the time-series format, and Non-IID financial data, and to propose an innovative deep matching approach for stock portfolio selection using deep stock profiles. For the majority of the quantitative fund managers, their task is not to achieve absolute returns, but to outperform a specific benchmark. The stock portfolios provided by TS-Deep-LtM algorithm can make them achieve 30% or 2%-15% higher annualized return than the CSI300 index or classical LtR approaches, respectively, during the entire backtestig period crossing bull and bear markets.

The rest of this paper is organized as follows. In the next section, we expound the deep stock profiling method and extensive related definitions, review the LtR algorithms, and introduce the deep matching approach. Section 3 describes the financial data sources, stock universe, and four different feature combinations of training data. Section 4 presents the labeling method and partitioning strategy for the training data, and evaluation metrics during two stages of experiments. Section 5 analyzes the experimental results of stock selection models and trading strategies and outlines an intelligent system framework for stock selection. The key findings of quantitative trading strategies based on stock portfolios and deep matching models are discussed and future research directions are proposed in Section 6. The main conclusions are summarized in Section 7.

## Methodology

### Deep stock profiling method

Exploring the factors that influence stock returns has always been the key to investigating the risky asset pricing problem [55]. From the perspective of modern asset pricing theory, centering on the mechanisms affecting stock price formation, as shown in Fig 1, we systematically extracted the stock features from multi-source heterogeneous financial big data with a deep stock profiling method. These features could be used to comprehensively depict the financial attributes of a risky asset.

Studies of modern asset pricing theory supported by standard and behavioral finance have specified many factors that affect cross-sectional stock returns, such as asset quality, asset growth, investor sentiment, and investor attention. Deep stock profiling was a collection of knowledge representation methods, which included the econometrics-based statistical formulas and deep-learning-based models. For example, we trained a financial Chinese text sentiment classification model using a deep learning algorithm and used it to extract sentiment representations of stocks from stock forum data.

**Weekly stock feature representation.** In this study, we chose to work with a weekly time granularity; namely, the first to last trading date in a calendar week, rather than a simple calendar week. As shown in Fig 2, stock profiles were extracted weekly and were converted into a "weekly stock feature vector" accordingly. Based on it, we proposed the "weekly stock feature

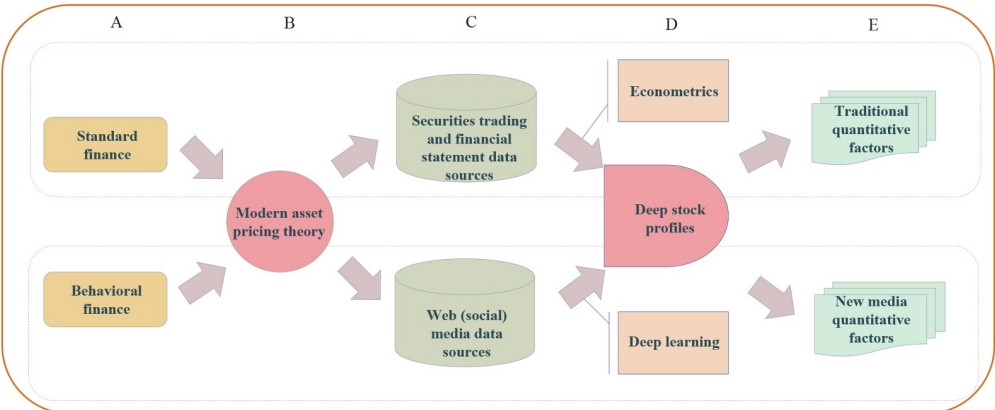

**Fig 1. Flow diagram of deep stock-profiling method.** As it is shown centering on the mechanisms affecting stock price formation in standard and behavioral finance (A), from the perspective of modern asset pricing theory (B), the stock features are extracted from two types of data sources (C) and are used for constructing deep stock profiles (D), which consisted of both traditional and new quantitative factors (E).

matrix" and "weekly stock selection target matrix" for the first time, rendering richer sequential information contained in the data objects. We defined a group of stocks in the stock universe as $S^{(t_i)} = \{s_1^{(t_i)}, s_2^{(t_i)}, ..., s_p^{(t_i)}\}$, and all weekly trading dates throughout the entire experiment period were expressed as $T = \{t_1, t_2, ..., t_n\}$.

Weekly stock feature vectors depicted the current financial characteristics of stocks in weekly trading windows, and they were represented as $1 \times m\text{-}dimensional$ vector. Starting from the current weekly trading date $t_i$, the corresponding stock feature vectors in backtracking $(w - 1)$ weekly trading dates were combined to form stock feature matrix $s_j^{(t_i)}$.

**Weekly stock selection target representation.** The stock selection target representation proposed in this study refered to the ideal state of investment returns for all stocks in the stock universe. Specifically, for stock $s_j$, starting from the weekly trading date $t_i$, the corresponding stock yields in weekly stock feature matrix $s_j^{(t_i)}$ were ranked in descending order. The corresponding feature vectors of the top k weekly trading dates were integrated as a representation

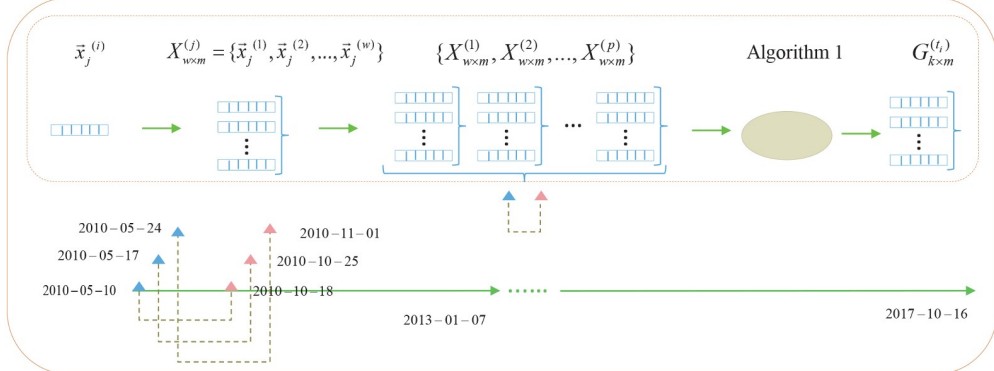

**Fig 2. Process of generating weekly stock feature vector and feature matrix.** The lower half in this figure shows the backtracking window, which is generated by time series. In the upper half of this figure, $\vec{x}_j^{(i)}$ and $X_{w \times m}^{(j)}$ represent weekly stock feature vector and feature matrix at the trading date $t_i$, respectively. $X_{w \times m}^{(1)}, X_{w \times m}^{(2)}, ..., X_{w \times m}^{(p)}$ and $G_{k \times m}^{(t_i)}$ represent all stock feature matrices in the stock universe and a stock selection target matrix generated by Algorithm 1 at the trading date $t_i$, respectively.

matrix under the optimal yield conditions (i.e., $s_j^{(t_i)}$). The specific implementation process is shown in Algorithm 1.

**Algorithm 1**: Representation algorithm of weekly stock selection target

```
Input: weekly trading date tᵢ, a group of stocks S = {s₁, s₂, ..., sₙ},
matrix representation X_{w×m}^{(j)} = {x⃗_j^{(1)}, x⃗_j^{(2)}, ..., x⃗_j^{(w)}} of stock sⱼ, historical
return set R_j^{(tᵢ)} = {r_j^{tᵢ−w+1}, r_j^{tᵢ−w+2}, ..., r_j^{tᵢ}} corresponding to stock sⱼ
Output: weekly stock selection traget representation matrix
G^{(tᵢ)} = {g⃗^{(1)}, g⃗^{(2)}, ..., g⃗^{(k)}}
1  w = a ← the backtracking window, a ∈ N+;
2  m = b ← the dimensions of stock feature vector, b ∈ N+;
3  k = c ← the weekly trading date window for the optimal yields, 1 < c
< = w;
4  n = e ← the size of a group of stocks, e ∈ N+;
5  d_f = 2 ← the first dimension reduction parameter;
6  d_s = m ← the second dimension reduction parameter;
7  D_s^{(tᵢ)} = {};
8  for i ← 0 to n do
9     sort R_j^{(tᵢ)} in descending order and adjust X_{w×m}^{(j)} according to the corre-
sponding position. The sorted matrix representation is recorded as
X_{w×m}^{(j)}′;
10    select the top-k vectors from X_{w×m}^{(j)}′ and generate a k × m-dimen-
sional matrix D_f^{(j)} = {v⃗_j^{(1)}, v⃗_j^{(2)}, ..., v⃗_j^{(k)}};
11    use algorithm t-SNE to reduce the dimension of the feature matrix
D_f^{(j)} to generate a k × d_f-dimensional matrix D_f^{(j)′} [56];
12    concatenate D_s^{(tᵢ)} and D_f^{(j)′} in the column direction and generate a new
feature matrix D_s^{(j)};
13 end for
14 use PCA model to reduce the dimension of the matrix D_s^{(tᵢ)} and generate
a k × d_s-dimensional matrix D_s^{(tᵢ)′} [57];
15 G_{k×m}^{(tᵢ)} = D_s^{(tᵢ)′};
16 return G_{k×m}^{(tᵢ)};
```

The critical steps of Algorithm 1 include screening and representing the optimal return yield states of stocks in the stock universe (step 10), and integrating corresponding matrix representations via manifold learning (step 11, 12, 14).

**Chinese text sentiment classification model in financial field.** To more accurately extract the sentiment features of stock profiles from financial social media texts, we customized the methods of word segmentation and labeling training data for financial Chinese texts. These methods were used to train a word embedding model and a text sentiment classification model based on deep learning (S1 File). Fig 3 shows the main modules of the text classification model framework.

We first started with the labeling of text sentiment and divided the social media texts of stock topics into three types of sentiment orientations: positive, neutral, and negative. A total of 120,000 texts were selected randomly from 30 million texts on stock topics in social media and were labeled. Out of these, 42,068, 59,095, and 18,837 texts were labeled as positive, neutral, and negative by professional annotators. The digits 1, 0, and -1 were used to represent positive, neutral, and negative investor sentiments in feature extraction. Then based on the text corpus collected from financial social media, we generated a professional word segmentation dictionary using manual extraction by experts. The dictionary included stock names, network terms (ghost stocks, penny stocks, junk stocks, etc.), jargon, vernacular terms, social media-specific emojis, main business of enterprises, financial report terms, etc. A total of 13,945

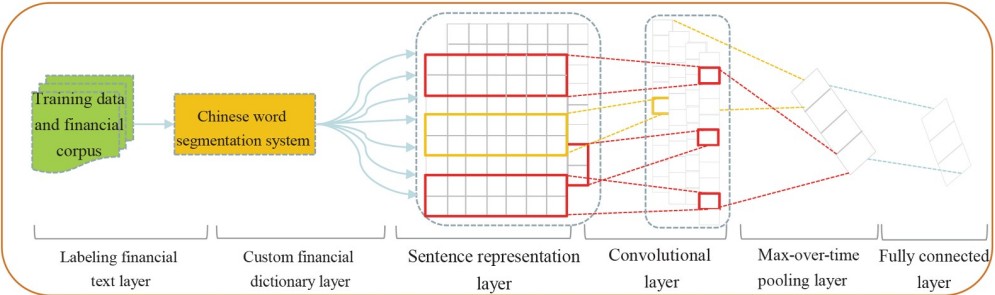

**Fig 3. Framework of Chinese text sentiment classification model based on CNN.** As it is shown the end-to-end framework is started with the labeling of financial texts, followed by custom financial Chinese text segmentation dictionary and main network structure of a deep CNN algorithm.

words in a custom dictionary were added into the Ansj [58] default general dictionary to perform Chinese text segmentation of stock topic discussions. Next, the word2vec algorithm [59] was used to train a word embedding model for text representation. A total of 83,979,599 posts and comments relating to 1000 stocks, most of which were under 100 words, were used as a training corpus. The output dimensionality of word vectors was 128, the minimum number of omits was 10, and the window was 3 in the word embedding model. Lastly, we adopted a deep CNN algorithm [60] to train the text sentiment classification model. The filter sizes were 3, 4, and 5; the number of filters was 128; the dropout was 0.5; the iterations were 200, and the batch size was 64. Under the method of cross validation, the 120,000 annotated data were divided into training, testing, and verification sets, with a model accuracy of 73% on the testing set. The posts and comments for each stock were counted by week. The calculation of investor sentiment for a stock at the i-th week was formulated as

$$P_{sentiment}^{(i)} = 1 \times N_{pos} + 0 \times N_{neu} + (-1) \times N_{neg}.$$

## Classical LtR algorithms

The nature of LtR is to solve ranking problems with classification or regression methods in machine learning [29]. Supervised LtR tasks (e.g., an information retrieval problem) are generally described as follows. Given a set of retrieval (query document pair) lists, each query $q_i \in Q$ contains a series of candidate documents $D_i = d_{i_1}, d_{i_2}, ..., d_{i_n}$ and they are labeled according to the relevance, propensity, or importance of the documents to a query item, such as two-level tagging (0 or 1) or multi-level tagging (0, 1, 2, 3, etc.). The LtR function is formulated as Eq (1):

$$h(w, \phi(q_i, D_i)) \Rightarrow R, \tag{1}$$

where $\phi(\cdot)$ is a feature representation function, which is used to map the relationship between a query item and a document. The weight indicator $w$ represents the set of parameters learned during the training of the LtR model.

According to different input and output spaces and loss functions, the LtR algorithms are divided into pointwise, pairwise, and listwise. The typical LtR framework consists of two modules: a learning system and a ranking system. The learning system mostly inludes some basic machine learning algorithms; the ranking system is responsible for setting loss function and controlling the optimization process. In previous research, the stock selection problem has been transformed into a ranking problem for a group of stocks and two ranking learning algorithms were used to train stock ranking prediction model for constructing trading strategies [28]. In this study, we first comparatively analysed the classical LtR algorithms in a broader

scope, including RankBoost [61], RankNet, MART [62], ListNet, AdaRank [63], RFRanker [64], Coordinate Ascent [65], and LambdaMART [66], to identify the optimal ranking prediction models. Then these models were used for a comparative analysis with our proposed deep matching method.

## TS-Deep-LtM algorithm

In this study, we translated the stock selection task into a matching problem between a group of stocks and a stock selection target. To deal with this problem, we devised a deep matching approach (TS-Deep-LtM) for stock portfolio selection based on deep text matching models [67–69]. The text matching problem and correlative algorithms are elaborated as follows.

Text matching has been placed in the core position of natural language processing (NLP) tasks [70]. Many NLP tasks can be abstracted as text matching problems [71–73]. For instance, information retrieval can be reduced to the matching between query items and documents. Question answering can be summed up to the matching between questions and candidate answers. Mathematically for a text matching task, given annotated training data set is $\mathbf{Z}_{train} = \{(z_1^{(i)}, z_2^{(i)}, r^i)\}_{i=1}^N$, where $z_1^{(i)} \in Z_1$ and $z_2^{(i)} \in Z_2$ are two paragraphs of text. If in a search engine problem, $z_1^{(i)} \in Z_1$ and $z_2^{(i)} \in Z_2$ are query items and documents, respectively. In a question-answering system, $z_1^{(i)} \in Z_1$ and $z_2^{(i)} \in Z_2$ are questions and answers, respectively. $r^i \in R$ represents the matching or correlation degree between $z_1^{(i)}$ and $z_2^{(i)}$. The objective of text matching is to automatically learn the matching model $f: Z_1 \times Z_2 \Rightarrow R$ on the training data. This allows any input $z_1^{(i)} \in Z_1$ and $z_2^{(i)} \in Z_2$ on the testing set $\mathbf{Z}_{test}$ to predict the matching degree $r^i$ between $z_1$ and $z_2$, thus obtaining the matching degrees ranking result.

As shown in Fig 4, the basic architecture of a deep text matching algorithm consists of a feature representation layer, a deep matching layer, and a pairwise ranking loss layer. The difference between different deep text matching models is largely determined by the matching architecture in the middle layer. Depending on the matching architecture, deep text matching models fall into two major categories: interaction-based models (e.g., DRMM and Conv-KNRM) and representation-based models (e.g., MV-LSTM). The deep text matching models used in this study included a deep relevance matching model (DRMM) [74], a convolution kernel-based neural ranking model (Conv-KNRM) [75], and a multi-view long short-term memory (MV-LSTM) [76] (S1 File). DRMM is a neural ranking model, which performs histogram pooling on the embedding based translation matrix, uses the binned soft-TF as a input of a feed-forward network, and calculates the similarity scores. KNRM adopts a novel kernel-pooling approach to softly count word matches at different similarity levels and to provide soft-TF ranking features. Conv-KNRM is a variant of KNRM [77], which applies a convolution to represent the n-gram embeddings of queries and documents. MV-LSTM is a neural semantic matching model, which obtains word embeddings by passing the sentences through a bidirectional LSTM algorithm (Bi-LSTMs) [78] and computes an interaction tensor between two embeddings using tensor layer.

TS-Deep-LtM integrated DRMM, Conv-KNRM, and MV-LSTM and aimed to deal with challenges posed by a low SNR, the time-series format itself, and Non-IID financial data. As shown in Fig 5, the algorithm architecture of TS-Deep-LtM mainly consisted of a model tuning module and a model selection module. The built-in parameter discovery mechanism in a model tuning module calculated the scores of the parameter combinations from the parameter space of each algorithm. The score set generated by all parameter combinations for the

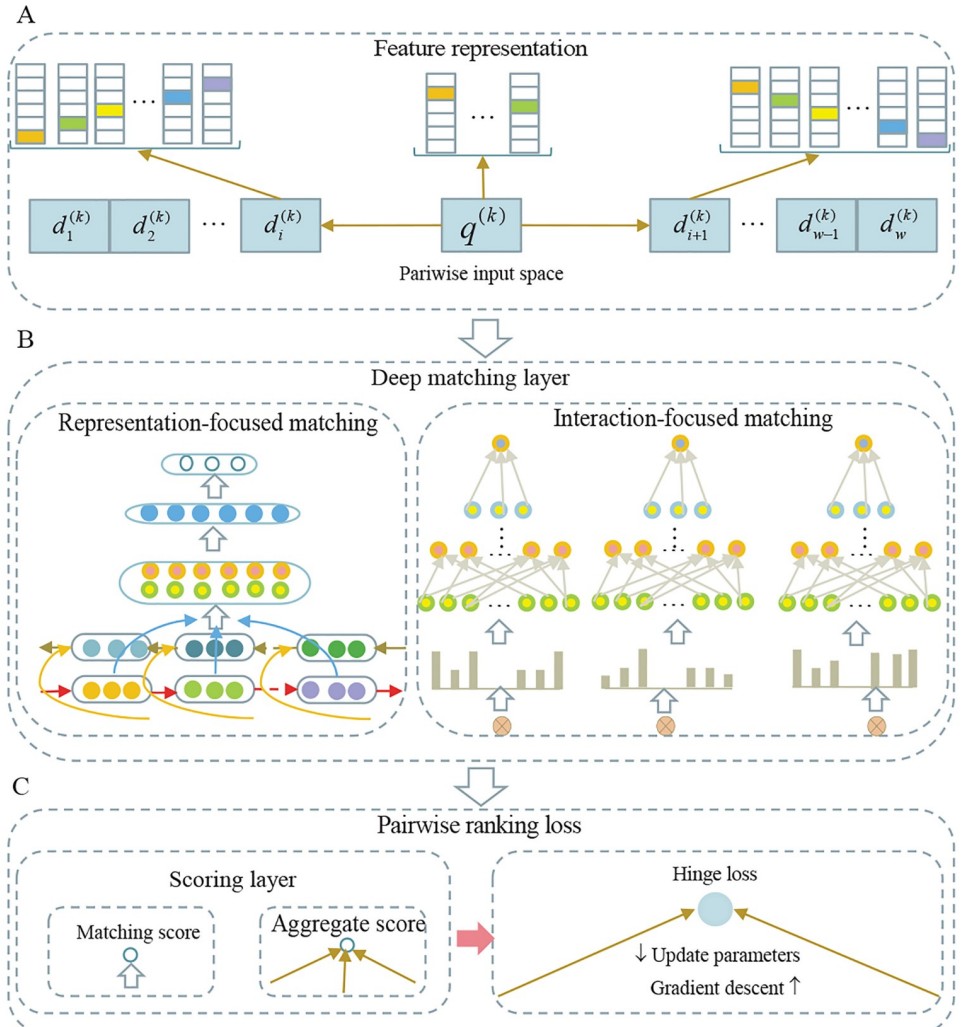

**Fig 4. Architecture of deep text matching algorithm.**

given algorithm $i$ was mathematically formulated as $A^{(i)} = \{s_1, s_2, \ldots, s_k\}$. The parameter combination corresponding to the highest score ($Max(A^{(i)})$) was taken as the optimal parameter combination for algorithm $i$. In model selection module, the built-in algorithm selection mechanism calculated the standard deviation $\sigma_{A^{(i)}}$ of score set $A^{(i)}$ and summarized the standard deviations of scores $B = \{\sigma_{A^{(1)}}, \sigma_{A^{(2)}}, \ldots, \sigma_{A^{(h)}}\}$ for all algorithms. In this study, we set $h = 3$ and calculated standard deviation $\sigma_B$ of all values in $B$. Then, we set $a = 0.01$ $and$ $b = 0.035$, and selected the models according to the method shown in Algorithm 2. The models were trained based on the selected algorithm and corresponding the optimal parameters.

**Algorithm 2**: Selection mechanism in the model selection module for TS-Deep-LtM

**Input:** standard deviation set of all algorithms $B = \{\sigma_{A^{(1)}}, \sigma_{A^{(2)}}, \ldots, \sigma_{A^{(h)}}\}$, score set $A^{(i)} = \{s_1, s_2, \ldots, s_k\}$ of algorithm $i$, parameter combination set of all algorithms $C = \{P^{(1)}, P^{(2)}, \ldots, P^{(h)}\}$, parameter combination set $P^{(i)} = \{p_1, p_2, \ldots, p_k\}$ of algorithm $i$

**Output:** the optimal parameter combination

1 $a, b \in (0, 1) \leftarrow$ thresholds for screening;
2 calculate the standard deviation $\sigma_B$ of $B = \{\sigma_{A^{(1)}}, \sigma_{A^{(2)}}, \ldots, \sigma_{A^{(h)}}\}$;

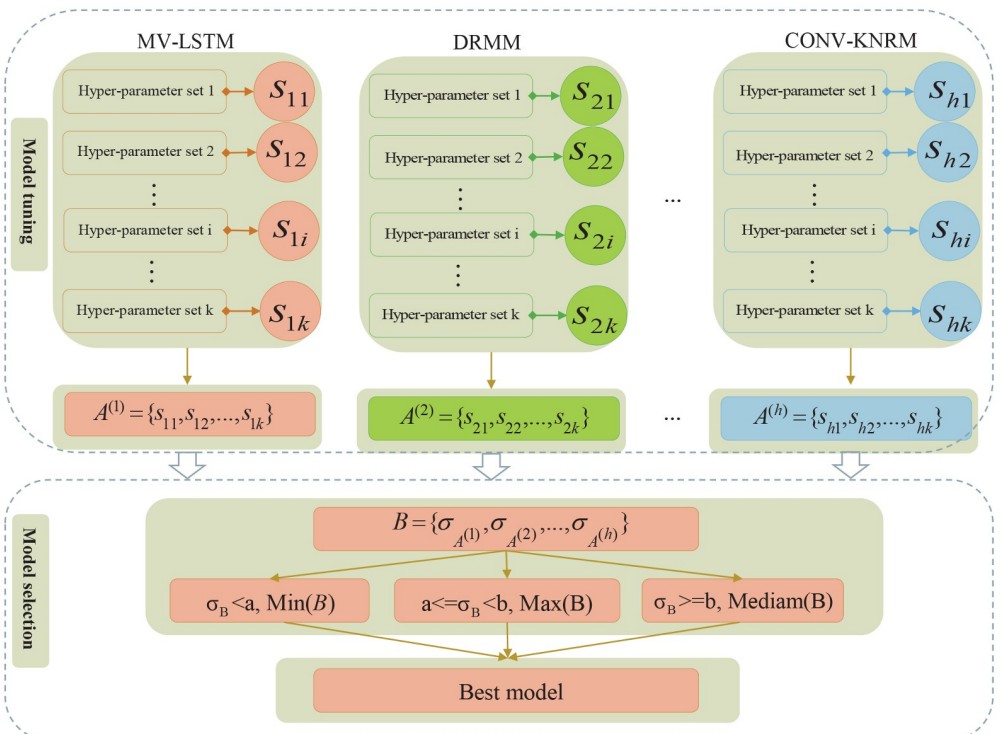

**Fig 5. TS-Deep-LtM algorithm design framework.** As it is shown the framework is composed of a model tuning module and a model selection module. $A^{(i)}$ denotes the set of scores for different parameter combinations of algorithm i. $B$ represents the set of different $\sigma_A^{(i)}$. The finally "Best model" refers to the selected algorithm and the optimal feature combination of the training data.

```
3 if σ_B < a then
4    according to σ_{A^{(o)}} = Min(B), select algorithm o;
5 end if
6 else if a < = σ_B < = b then
7    according to σ_{A^{(o)}} = Max(B), select algorithm o;
8 end if
9 else
10    according to σ_{A^{(o)}} = Median(B), select algorithm o;
11 end if
12 according to the sequence number corresponding to s_f = Max(A^{(o)}),
further select the optimal parameter combination p_f from P^{(o)};
13 return p_f^{(o)};
```

We used the implementation components of deep text matching algorithms provided by MatchZoo [79] to design TS-Deep-LtM alogrithm. In the process of model pre-training, the hyper-parameters involving in optimization included optimizer, the layers of the multilayer perceptron, the distribution of units in each layer, activation function, dropout rate, and learning rate. Out of these, the optional parameter values for optimizer included Adam, Adagrad, and RMSProp [80]. The parameter values for the layers and distribution of units were obtained from the quantitative uniform distribution. In addition, some hyper-parameters peculiar to different deep text matching algorithms, such as the number of lstm units for MV-LSTM, the number of filters, max n-gram, and convolution activation function for Conv-KNRM, and k-max pooling for DRMM, were included.

### Stock portfolios based on ranking or matching models

The prediction results of the stock ranking models or stock deep matching models were the correlations between a group of stocks and rankings or matching levels between a group of stocks and stock selection targets, respectively. According to the correlations or matching degrees, this group of stocks were ranked in descending order (S2 and S3 Files), and the ranking result was used as a stock selection signal. Specifically, during the entire backtesting period, we selected a stock portfolio according to a frequency of weekly position adjustments. On each position adjustment day, according to the prediction results of the model, we bought and held the top 40 stocks as a portfolio and sold out on the next position adjustment day.

## Data descriptions

### Data resources

The sources of financial data used in this study consisted of three major categories: 1) the stock trading data of China's Shanghai and Shenzhen stock markets from Wind Financial Terminal (one of China's most senior finance information service providers; https://www.wind.com.cn/), 2) the quantitative investment data from DATAYES (one of China's most professional financial big data providers; https://m.datayes.com/), and 3) the user-generated content (UGC) on financial social media from a stock themed community owned by East Money Information (one of the most visited and influential financial securities portals in China; http://www.eastmoney.com/). The stock trading data provided data support for us to screen a stock universe. The quantitative investment data was the basis for generating traditional quantitative factors used in this study. The new social media quantitative factors were extracted from financial social media data. Additionally, the backtesting experiments in this study were conducted using the trading environment simulated by UQER (https://uqer.io/).

### Stock universe

We defined a stock universe in accordance with three steps of screening: age of stocks (listing date and continuity state), liquidity of stocks (turnover rate), and number of posts on stock forums per week. We first screened a total of 1810 stocks, which were first listed before May 2010 and continuously existed until October 2017, from the Shanghai and Shenzhen stock markets. Then, the 1810 stocks were ranked in descending order according to the weekly average turnover rate, and the top 1000 stocks with the highest liquidity were selected. The stocks with zero weekly postings and frequencies greater than 7 times were further eliminated, which left a total of 314 stocks that were kept as our final stock universe.

The distribution of 314 stocks in 10 first-level China Securities Index (CSI) industry classification fields (http://www.sse.com.cn/assortment/stock/areatrade/ahassortment/) is shown in Fig 6. Compared with the industrial distribution of the constituent stocks in the CSI300, our stock universe leaned towards large-cap constituent stocks.

### Feature combinations

The features of training data were derived from stock profiles with traditional and new factors. For each factor, we selected several of the most representative variables [35–37, 43–49] that had been empirically tested by multi-factor models [5]. These proxy variables for each factor together constituted the stock feature combination.

**Feature combination of traditional factors.** High-quality traditional quantitative factors after repeated validation can be obtained from professional quantitative investment platforms. This study screened eight groups of quantitative factors from a quantitative factors library

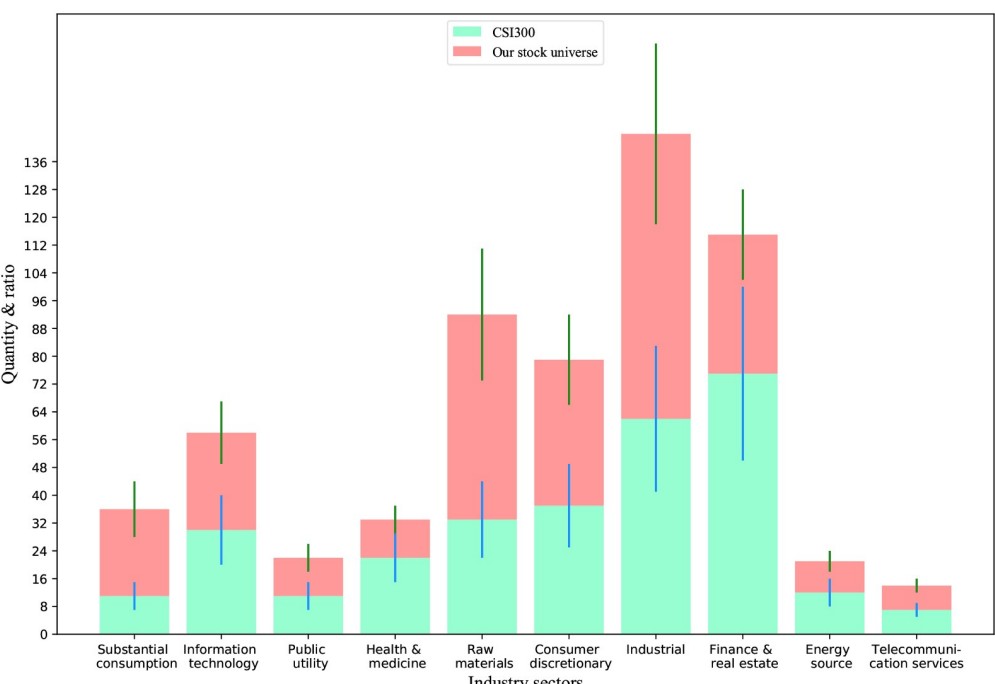

**Fig 6. Distribution of stocks by China securities regulatory commission industries.** The stacked bar chart represents the number of stocks in an industry, and short line on the bar indicates the ratio of the number of stocks to corresponding the total number of constituent stocks.

(with more than 400 factors) provided by UQER (https://uqer.io/). These eight groups of quantitative factors covered classical factors which had been reliably demonstrated by modern asset pricing theory and empirical analyses. The specific types and their detailed descriptions are summarized in Table 1. These factors consisted of a total of 18 weekly indicators, which were an important part of stock profiles and formed the feature combination of training data for comparative experiments in this study.

**Feature combination of social media factors.** For emerging social media quantitative factors, there is no generally accepted the sources of social media data and the standard of factors extraction. This study collected UGC data from the two dimensions of posts and user accounts. From the dimension of posts, the full history posts for each stock, including post text, post publisher, publication date, terminal of publishing posts, the number of read, and commented posts, were collected. From the dimension of user accounts, the registration time, the number of followers, the number of following accounts, etc., for each account were accessed. Finally, the UGC data representing 1000 stocks from 29,881,168 posts and 1,686,203 user accounts were collected. As shown in Table 2, we extracted four groups of factors from UGC data with a stock profiling method. These social media factors consisted of a total of seven weekly indicators, which were used to form the feature combination of training data for comparative experiments in this study.

**Feature combination of profile factors.** The stock profile quantitative factors consisted of 18 traditional factors and seven social media factors and formed the feature combination of training data for comparative experiments in this study (S4 File).

**Feature combination of Song et al. factors.** The baseline feature combination was composed of six quantitative factors in two categories (Table 3). Out of these, the sentiment shock

**Table 1. Feature combination from traditional quantitative factors in deep stock profiling.**

| Types of factors | Proxy indicators of factors |
|---|---|
| Asset quality factor | Return on assets |
| | Quick ratio |
| Asset value factor | Price-earnings ratio |
| | Price-to-book ratio |
| | Price-to-cash-flow ratio |
| | Price-to-sales ratio |
| Asset growth factor | Total assets growth rate |
| | Operating profit ratio |
| Asset size factor | Natural logarithm of total market values |
| Momentum factor | Momentum index |
| | 1-week leading return |
| | 1-month leading return |
| Market factor | Historical daily beta |
| Technical factor | Relative strength index |
| | Accumulation swing index |
| | Moving average convergence divergence |
| Emotion factor | Psychological line index |
| | Volatility of daily turnover during the last N days |

All quantitative factors are extracted according to the time window of trading week and the time span of each trading week ranges from the first trading day to the last trading day.

and trend factors were two derived emotional factors that were extracted from raw emotional factors [81], with the method of feature abstraction being the same as Song et al. [28].

## Experiment design

### Training data

A weekly stock selection target representation and the weekly stock feature representation in the stock universe constituted a group of training data (S4 File). For the LtR algorithms, the input forms of training data were weekly stock feature vectors (Fig 7B). For the TS-Deep-LtM algorithm, the input forms of training data were weekly stock feature matrices (Fig 7C) and weekly stock selection target matrices (Fig 7D). The preprocessing of training data we

**Table 2. Feature combination from new social quantitative factors in deep stock profiling.**

| Types of factors | Proxy indicators of factors |
|---|---|
| Investor sentiment factor | Sentiment polarity value of users' posts weekly |
| Investor network interactive factor | Information acquisition capabilities, like the number of followers etc. |
| | Information dissemination capabilities, like the number of followers etc. |
| Investor attention factor | Number of users' posts weekly |
| | Reading volume of users' posts weekly |
| Investor maturity factor | Investment experience, like social account registration time |
| | Investment specialization, like release terminal and posting time |

All quantitative factors are extracted according to the weekly time window, which refers to the period between the opening of the first trading day last week and the opening of the first trading day this week.

**Table 3. Baseline feature combination.**

| Types of factors | Proxy indicators of factors |
|---|---|
| Investor sentiment factor | sentiment shock score |
| | sentiment trend score |
| | 1-week leading average sentiment |
| | 1-month leading average sentiment |
| Momentum factor | 1-week leading return |
| | 1-month leading return |

The investor sentiment feature group includes four derived investor sentiment factors, which are calculated based on the investor sentiment factor extracted in this study.

performed included winsorize, missing data imputation, neutralization (industry neutralization and market value neutralization), and standardization, with the details shown in Fig 8.

## Labeling training data

Each stock matching pair (a stock and stock selection target pair) was given a label, which represented the matching level between a stock and a stock selection target (S5 File). The 314 stocks in the stock universe were ranked according to their corresponding weekly yields per trading date in descending order. Accordingly, the sequence was divided into four equal parts and labeled with four matching levels (i.e., 3, 2, 1, and 0).

## Dividing training data

This study adopted two ways of using rolling windows to divide the training data: by year versus by samples. The backtesting interval of trading strategies was from January 2013 to

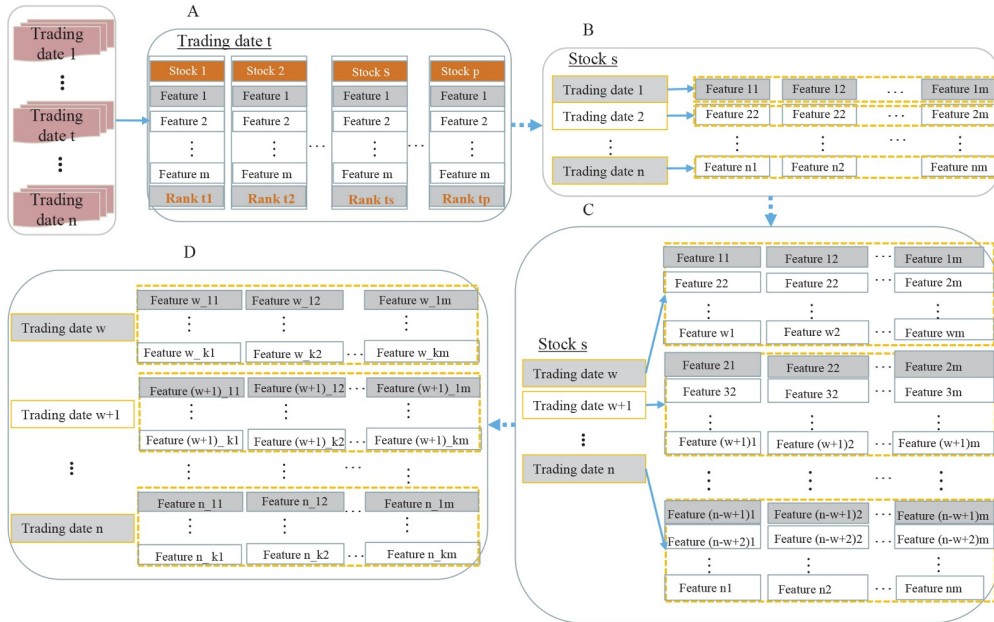

**Fig 7. Weekly stock feature representation and weekly stock selection target representation.** (A) denotes the stock profiles and yield rankings of all stocks on trading date $t_i$. (B) denotes the feature vectors of stock $s_j$ on weekly trading day series. (C) and (D) represent the feature matrices of stock $s_j$ on weekly trading date series and stock selection target matrices on trading date $t_i$, respectively. The parameters in this study were set to $m = 25$, $w = 24$, $k = 10$.

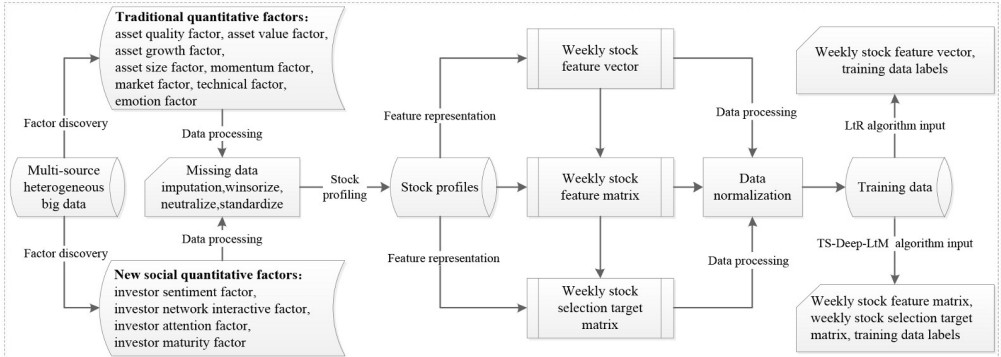

**Fig 8. Flowchart of feature extraction and data preprocessing.** From left to right, data preprocessing is performed around three modules including multi-source heterogeneous big data, stock profiles, and training data.

**Table 4. Dividing the training set, validation set, and testing set by year.**

| Year | Taining set | validation set | Testing set |
|------|-------------|----------------|-------------|
| 2013 | 2010.05-2011.12 | 2012.01-2012.12 | 2013.01-2013.12 |
| 2014 | 2011.01-2012.12 | 2013.01-2013.12 | 2014.01-2014.12 |
| 2015 | 2012.01-2013.12 | 2014.01-2014.12 | 2015.01-2015.12 |
| 2016 | 2013.01-2014.12 | 2015.01-2015.12 | 2016.01-2016.12 |
| 2017 | 2014.01-2015.12 | 2016.01-2016.12 | 2017.01-2017.10 |

October 2017. The dividing method by year is shown in Table 4. The dividing method by samples is shown in Fig 9. A total of 246 groups of training data were divided in training, validation, and testing sets in the entire sample period.

## Automatic trading system

On the UQER (https://uqer.io/) backtesting platform, we set an initial principal of RMB 10 million and adjusted the portfolio on a weekly frequency, with the backtesting interval ranging from January 2013 to October 2017. On each rebalancing day of portfolio, we bought 40 stocks

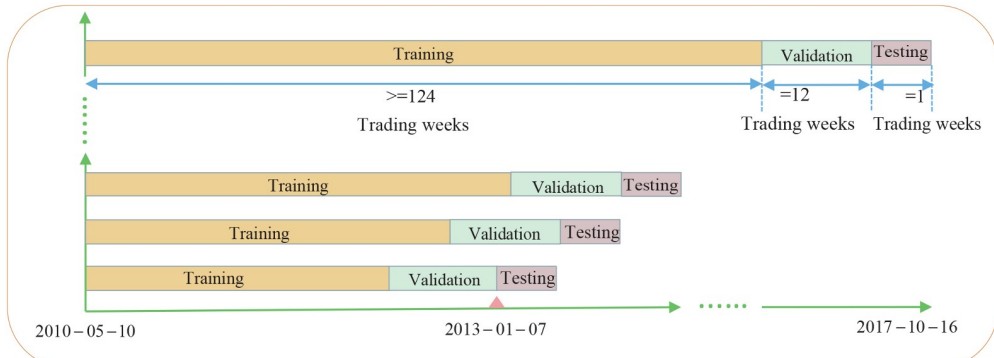

**Fig 9. Dividing the training set, validation set, and testing set by samples, throughout the sample period.** As it is shown starting from the last sample in 2012, a new sample set is obtained for each additional sample by a rolling window in turn, with the last sample in each sample set as the testing set, the backward 12 samples starting from the last sample as the validation set, and the remaining samples as the training set.

with equal weight for each one and meanwhile sold out previous 40 stocks bought on last position adjustment day. Then buying and selling out, in turn, until to the last day of the backtesting interval. Throughout the backtesting period, the position adjustment records of the trading strategies based on TS-Deep-LtM models is shown in S6 File. Finally, the evaluation metrics for the trading strategy were calculated.

## Performance measurement

**Evaluation metrics of ranking and matching problems.** For more than two correlation or matching levels for measuring a ranking result, the expected reciprocal rank [82] and normalized discounted cumulative gain (NDCG) [83] evaluation metrics are applicable [84]. Considering our four correlation or matching levels and the evaluation metrics used in similar studies, we used NDCG to evaluate models.

NDCG introduces the use of location information, with both the correlation levels of documents and their corresponding location information being considered. For the corresponding matching or correlation levels for the documents at each position, given that the optimal ranking is $\widehat{rel}_1, \widehat{rel}_2, ..., \widehat{rel}_n$ and the predicted ranking is $rel_1, rel_2, ..., rel_n$, the NDCG evaluation metric can be calculated using Eqs (2)–(4).

$$DCG@p = \sum_{i=1}^{p} \frac{2^{rel_i} - 1}{log_2(i+1)},$$ (2)

$$IDCG@p = \sum_{i=1}^{p} \frac{2^{\widehat{rel}_i} - 1}{log_2(i+1)},$$ (3)

$$NDCG@p = \frac{DCG@p}{IDCG@p}$$ (4)

The $rel_i$ in Eq (2) represents the yields produced from the document at the i-th position. Eq (2) promotes the affect proportion of correlation or matching levels. The IDCG in Eq (4) equals DCG in the ideal case. The NDCG in the Eq (4) represents normalization processing with IDCG, and indicates the gap between DCG and IDCG. The $p$ in the formulas was set to 10 in this study. During the process of model training, we evaluated the generalization ability of models with training loss and NDCG@10.

**Evaluation metrics of trading strategies in backtesting experiment.** This study used CSI300 as the benchmark strategy. In general, strategies with higher returns than those of the benchmark strategy are preferred. Positive metrics (returns, annualized return, alpha, Sharpe ratio, and information ratio), negative metrics (beta, volatility, and max drawdown), and turnover rate were used to evaluate trading strategies [85–87].

- **Annualized return** refers to the return obtained within a one-year investment period, which is obtained by converting the current return (i.e. daily return, weekly return, and monthly return).

- **Sharpe ratio** is used to evaluate risk-adjusted performance. A large and positive Sharpe ratio suggests a stock risk-adjusted overperformance and a low and negative ratio indicates underperformance. It is formulated as $SharpeRatio = (R_p - R_f)/\sigma_p$, where $R_p$ is the return on trading strategy $p$, $R_f$ represents the risk-free rate, and $\sigma_p$ represents the standard deviation of the trading strategy.

- **Alpha ($\alpha$)** is used to measure the excess return compared with a suitable market index. A positive alpha means the return of a trading strategy outperforms the market during that same period, and a negative alpha means the investment underperforms the market.

- **Beta ($\beta$)** is used to measure the sensitivity of the investment performance of a trading strategy to the market. A beta below 1 indicates that the volatility of a trading strategy is below that of the market.

- **Volatility ($\sigma$)** is used to measure the investment risk. The greater the volatility of a trading strategy, the higher the risk. It is formulated as $Volatility = \sqrt{[250/(n-1)] \sum_n^{i=1} (r_i - \bar{r})}$, where $r_i, \bar{r}, n$ are the daily return, the average return, and the number of days of executing a trading strategy, respectively.

- **Information ratio** is used to measure the excess return of a trading strategy per a unit excess risk. It is formulated as $InformationRatio = (R_p - R_f)/\sigma_t$, where $R_p - R_f$ is the excess return of a trading strategy, and $\sigma_t$ is the annualized standard deviation of the daily return difference between a trading strategy and the benchmark.

- **Maximum drawdown** refers to the maximum percentage drop incurred from a peak to a bottom in a certain time period. It is formulated as $Max\ Drawdown = \max_{0 \le t \le T}(\max_{0 \le x \le t} S_x - S_t)$, where $\max_{0 \le x \le t} S_x - S_t$ is the drawdown from the previous maximum value at time $t$.

- **Turnover rate** measures the trading activity during a particular period. Portfolios with high turnover rates usually produce high transaction costs. Generally, if the securities market is experiencing an upturn, investment returns will far exceed than the transaction costs and a high turnover rate is favorable. Conversely, if the securities market is experiencing a downturn, a low turnover rate is favorable.

These metrics were used to evaluate comprehensively the performance of trading strategies from the perspectives of stability, trading frequency, return, and risk. The volatility and turnover rate were used to evaluate the performance of stability and trading frequency, respectively. The evaluation of return performance adopted annualized return and alpha. Annualized return was used to measure the expected return of a trading strategy; alpha focused on measuring the excess return generated by active investing. The evaluation of risk performance adopted beta and max drawdown [88]. Beta was used to measure the systemic risk taken by a trading strategy; maximum drawdown focused on measuring the "worst-case" that might occur with a trading strategy. Sharpe ratio and information ratio were used to evaluate the performance of both return and risk. The key difference between the two is the definition of "excess return". Sharpe ratio defines excess return as the return above the risk free rate, thus it is a measure of absolute return and total risk. Information ratio defines excess return as the return in excess of a relevant benchmark index, thus it is a measure of risk-adjusted return by active investing [89]. Whatever the definition, the higher the Sharpe ratio (and information ratio) quotient the better.

## Results

The final assessment of experimental results involved the evaluations of the predictive performance of the stock ranking or matching models and the profitability of trading strategies. Through these two stages of evaluation, we chose the optimal feature combination, dividing method for the training data, and ranking prediction model, and demonstrated the superiority of the deep matching model by comparative evaluation. Based on the quantitative investment

approach and experimental findings presented in this study, we further designed an applicable system framework for intelligent investment.

## Ranking models and trading strategies based on LtR algorithms

For dividing the training set by year, five training sets were generated by rolling window in turn. We adopted classical LtR algorithms to train ranking prediction models on five training datasets (rolling window by year) with different feature combinations. Fig 10A–10D shows the predictive performances of ranking models in NDCG@10, on four different feature combinations. The ranking algorithms with higher NDCG@10 values included RFRanker, Coordinate Ascent, MART, RankNet, and RankBoost. Of these, the models trained based on the Rank-Boost algorithm had high NDCG@10 values, but they fluctuated significantly. Thus, the final performance of such models needed to be further evaluated in backtesting experiments.

As shown in Fig 10E–10I, the optimal performances in NDCG@10 were obtained when adopting the datasets (rolling window by year) with profile feature combination for the Coordinate Ascent, MART, and RankNet algorithms. A sub-optimal performance in NDCG@10 was obtained for the RFRanker algorithm. This preliminary result demonstrated the feasibility and effectiveness of our stock profile feature combination method in addressing the ranking problem. The models trained based on the RFRanker algorithm obtained higher NDCG@10 values overall on four different feature combinations compared with the other four models.

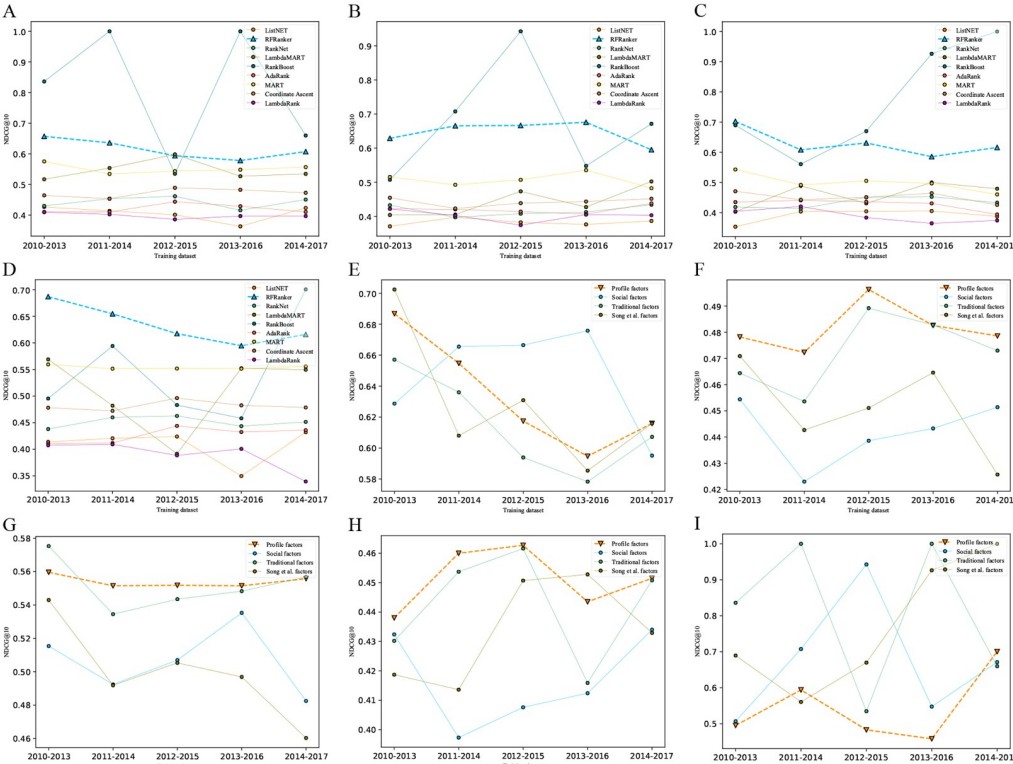

**Fig 10. Performances of different ranking models trained on different feature combinations.** (A)-(D) represents the performances of models trained with nine classical LtR algorithms on four different feature combinations. The feature combinations corresponding to (A)-(D) are traditional factors, social factors, Song et al. factors, and profile factors, respectively. (E)-(I) represents the performances of models trained on four feature combinations using five algorithms. The algorithms corresponding to (E)-(I) are RFRanker, Coordinate Ascent, MART, RankNet, and RankBoost, respectively.

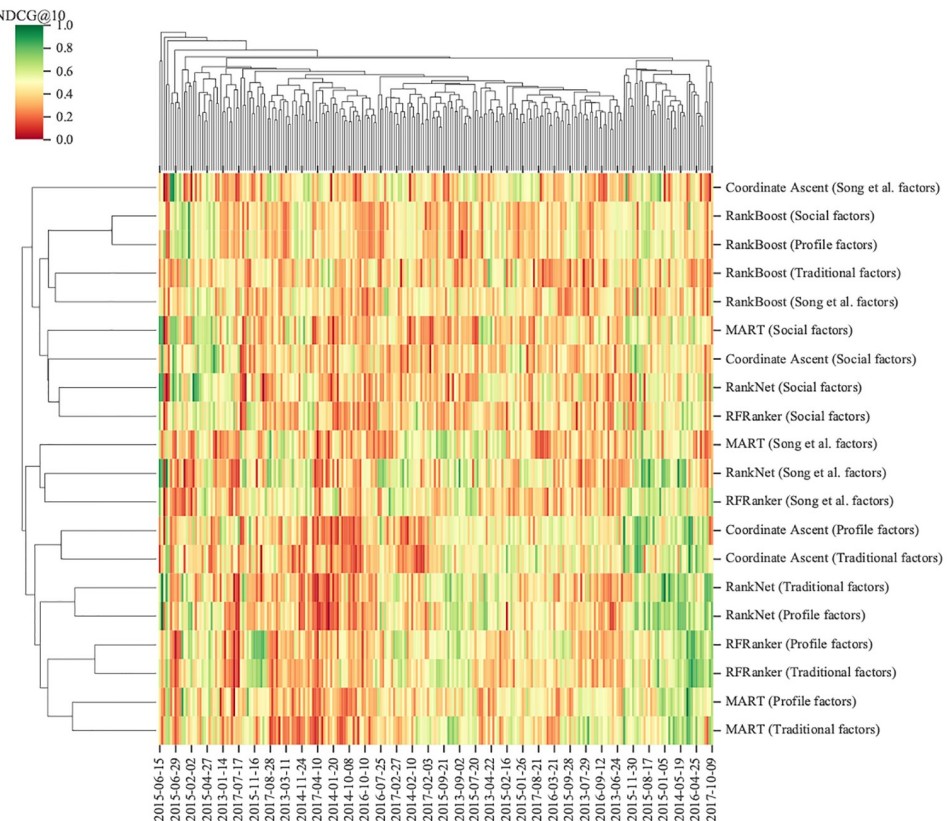

**Fig 11. Performances of models trained on four feature combinations using five classical LtR algorithms.** The different colors in figure represent corresponding NDCG@10 values. The right side of figure lists algorithms and feature combinations used in model training. The left side of figure shows corresponding clustering result of model performances. The models with similar performances are grouped under the same cluster.

For dividing the training set by samples, a total of 246 training sets were generated by a rolling window in turn. Then we adopted the aforementioned five algorithms (i.e., RFRanker, Coordinate Ascent, MART, RankNet, and RankBoost) to train ranking prediction models on the 246 training datasets with four feature combinations. Their NDCG@10 performances are shown in Fig 11. According to the clustering of NDCG@10, the models that were trained based on RFRanker, Coordinate Ascent, MART, and RankNet on the feature combinations of profile and traditional factors had superior performances. The predictive performances of the models trained based on RankBoost were highly variable, which was consistent with the result obtained in Fig 10A–10D.

After evaluating the ranking prediction models, we constructed trading strategies based on the prediction results of these models and evaluated their investment performances in backtesting experiments. As shown in Fig 12A and 12B, the trading strategies constructed by the models that were trained based on RFRanker, Coordinate Ascent, MART, and RankNet on the training datasets of rolling window by samples with profile feature combination had better investment performances in both annualized return and Sharpe ratios. This result was consistent with those in the evaluation of ranking prediction models. Of these, the trading strategies based on RFRanker algorithm on the training datasets of rolling window by samples with profile feature combination achieved the best annualized return (39%) and Sharpe ratio (112%) in the backtesting experiments.

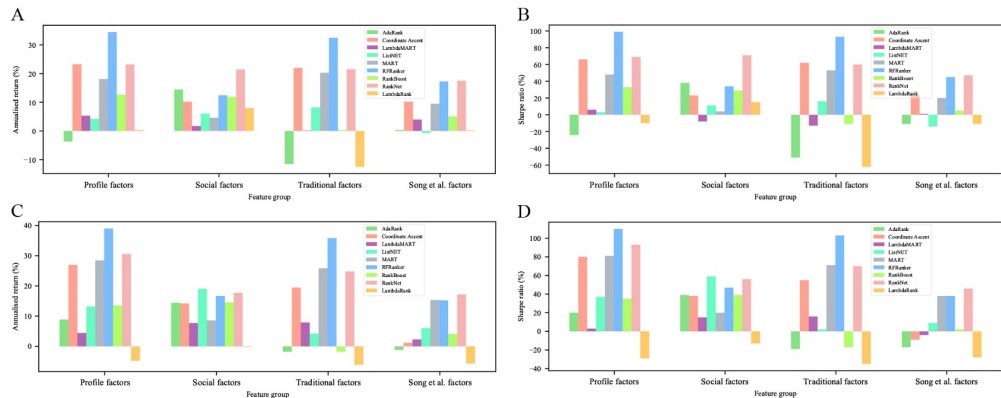

**Fig 12. Performances of different trading strategies by annualized indicators.** ((A) and (B)) and ((C) and (D)) represent the performances of trading strategies constructed by the models that were trained on the training datasets of rolling window by year and by samples, respectively, in both annualized return and Sharpe ratios. The legend in each chart shows the algorithms that are used to train ranking models and construct trading strategies.

Further, we conducted a more detailed comparison among the four types of trading strategies with superior investment performances. Fig 13 shows the performances of trading strategies in eight evaluation metrics based on different ranking models during the entire back-testing period. The trading strategies based on RFRanker with feature combinations of profile and traditional factors had superior performances in cumulative returns (Fig 13A). The trading strategies based on four algorithms with profile feature combinations obtained better drawdown overall, with RFRanker having the best performance (Fig 13B). As shown in Fig 13C, looking at obtaining positive returns for most trading strategies, those based on RFRanker earned higher returns. Conversely, in terms of suffering losses for most trading strategies, those based on RFRanker achieved fewer losses. The trading strategies based on RFRanker with profile feature combination obtained better performances in positive metrics, including Sharpe ratio, information ratio, and alfa (Fig 13D).

## Deep matching models and trading strategies based on TS-Deep-LtM algorithm

Inspired by the above experimental findings, we used the TS-Deep-LtM algorithm to construct deep matching models and trading strategies using the training data divided by samples with stock profile feature combination. During TS-Deep-LtM model training, the optimal parameter combinations were generated by the model tuning module, with the details shown in S7 File. As shown in Fig 14A, along with the increasing of iteration times, the training loss decreased gradually and then became steady. The NDCG@10 curves on the verification and testing sets first fluctuated significantly and thenmaintained relative stable. The difference between the verification set and the testing set in NDCG@10 first fluctuated wildly and then tended to be relatively stable. For the models with a comparable average predictive level, the model's final application performance could be determined by the model's volatility. As shown in Fig 14B, the models trained based on TS-Deep-LtM had a low standard deviation in NDCG@10 ($\delta = 0.1413$), indicating a more robust prediction ability than the RFRanker algorithm.

In Fig 15, the trading strategies constructed by the predictive results of TS-Deep-LtM were superior to those constructed by RFRanker in seven evaluation metrics, reflecting investment performances. Specifically, TS-Deep-LtM had higher positive evaluation metrics (i.e.,

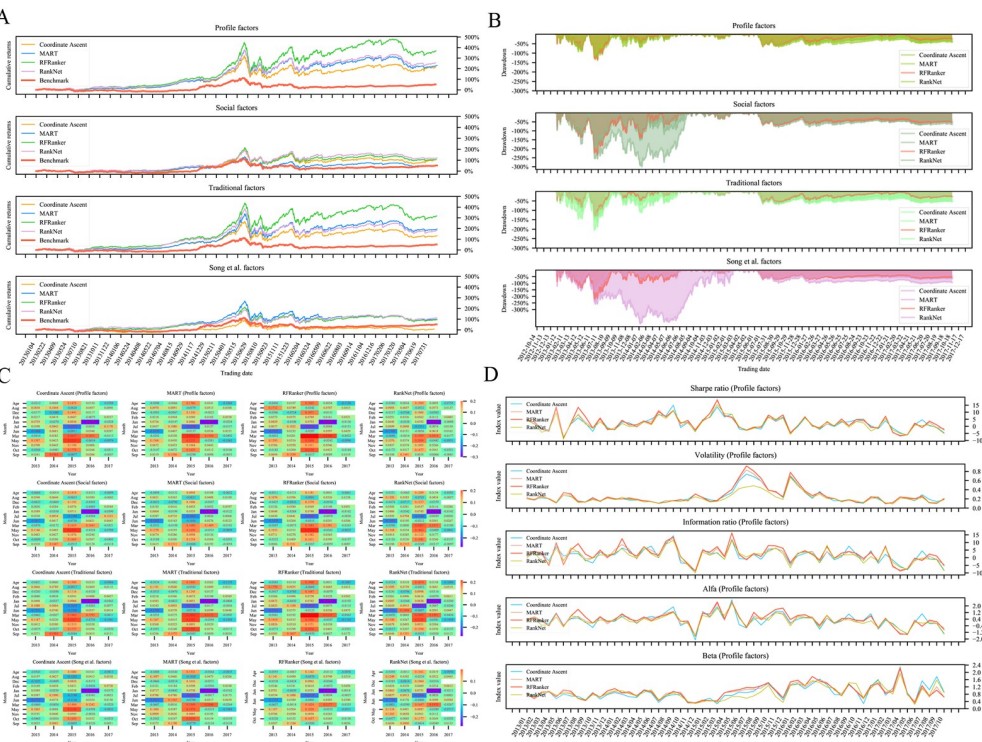

**Fig 13. Performances of different trading strategies in cumulative returns (A), drawdown (B), monthly returns (C), Sharpe ratio, volatility, information ratio, alfa, and beta (D) during the entire backtesting period.**

annualized return, alpha, Sharpe ratio, and information ratio) and lower negative evaluation metrics (i.e., beta, max drawdown, and volatility), compared to RFRanker. TS-Deep-LtM had a higher turnover rate, which was more favorable during periods of rising securities markets. The application performances of five models are listed in detail in Table 5, which shows the

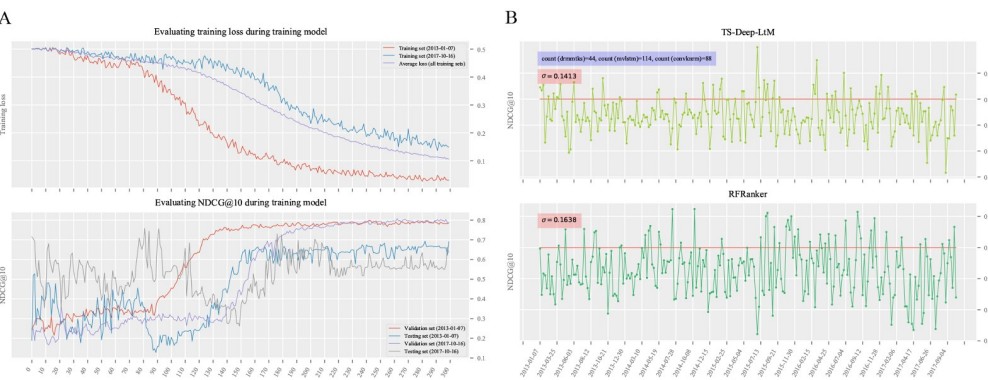

**Fig 14. Performances of TS-Deep-LtM models.** (A) evaluates the generalization ability of TS-Deep-LtM models. The trading date in (A) corresponds to a group of training data (training set, validation set, and testing set). We select the first and last set of training data to illustrate the training loss and NDCG@10 during model training. The average loss represents the mean of training losses on all training sets. (B) shows the performances of models trained on 246 training data sets based on TS-Deep-LtM and RFRanker algorithms during the entire sample period. The standard deviation $\sigma$ is used to evaluation the dispersion degree of 246 NDCG@10 values for each type of model. Among the models trained based on TS-Deep-LtM algorithm, DRMM-tks, MV-LSTM, and CONV-KNRM are selected for 44, 114, and 88 times, respectively.

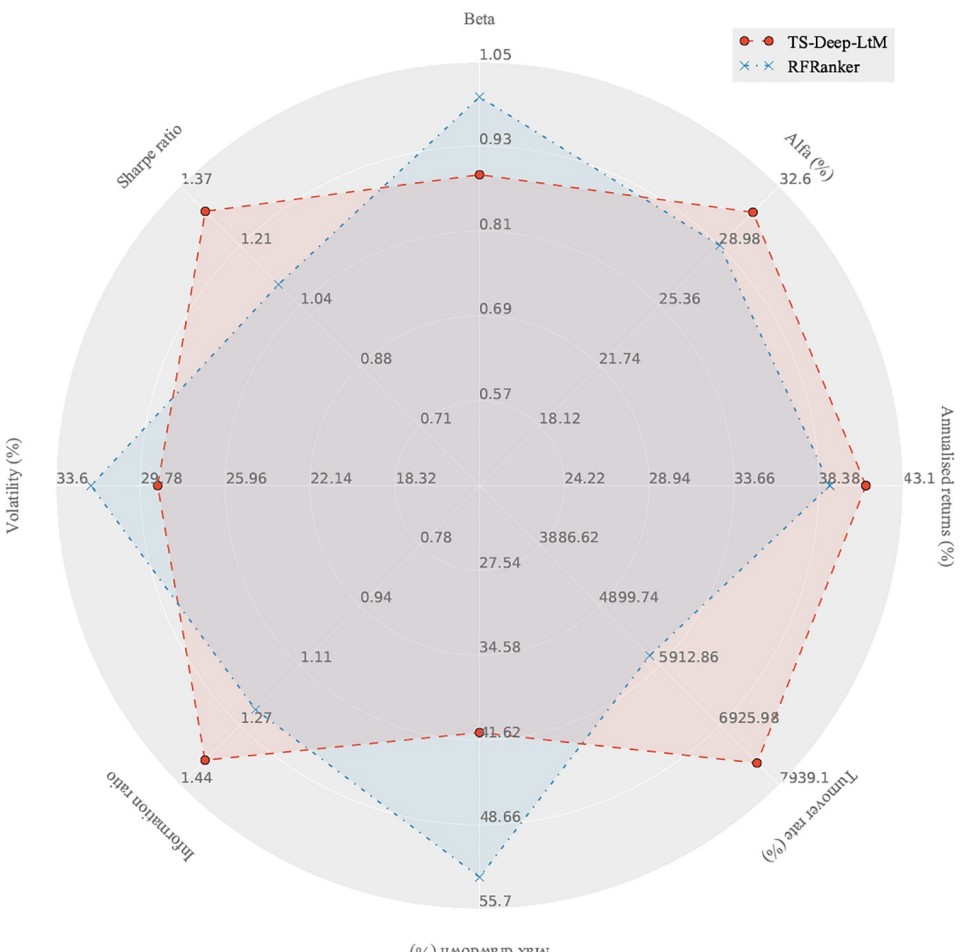

**Fig 15. Overall performance of trading strategies based on TS-Deep-LtM and RFRanker on the eight evaluation metrics.**

superiority of the deep matching model. Our method showed annualized return of 41.19% over the entire backtesting period, compared to 39% for the RFRanker method. For more information about the daily or monthly evaluation metrics of trading strategies, see S8–S15 Files.

**Table 5. Evaluation of long-only trading strategies based on different ranking prediction models.**

| Algorithm | Annualized return | Alpha | Beta | Sharpe ratio | Volatility | Information ratio | Max drawdown | Turnover rate |
|---|---|---|---|---|---|---|---|---|
| MART | 0.2843 | 0.1885 | 0.94 | 0.81 | 0.3060 | 0.87 | 0.4898 | 49.21 |
| Coordinate Ascent | 0.2696 | 0.1784 | 0.87 | 0.80 | 0.2943 | 0.77 | 0.5151 | **47.22** |
| RankNet | 0.3060 | 0.2154 | **0.86** | 0.93 | **0.2902** | 0.92 | 0.5066 | 55.09 |
| RFRanker | 0.3900 | 0.2906 | 1.00 | 1.10 | 0.3241 | 1.23 | 0.5385 | 57.47 |
| TS-Deep-LtM | **0.4119** | **0.3197** | 0.89 | **1.30** | 0.2903 | **1.37** | **0.4157** | 75.61 |

The best performing values for each metric in the table are shown in bold.

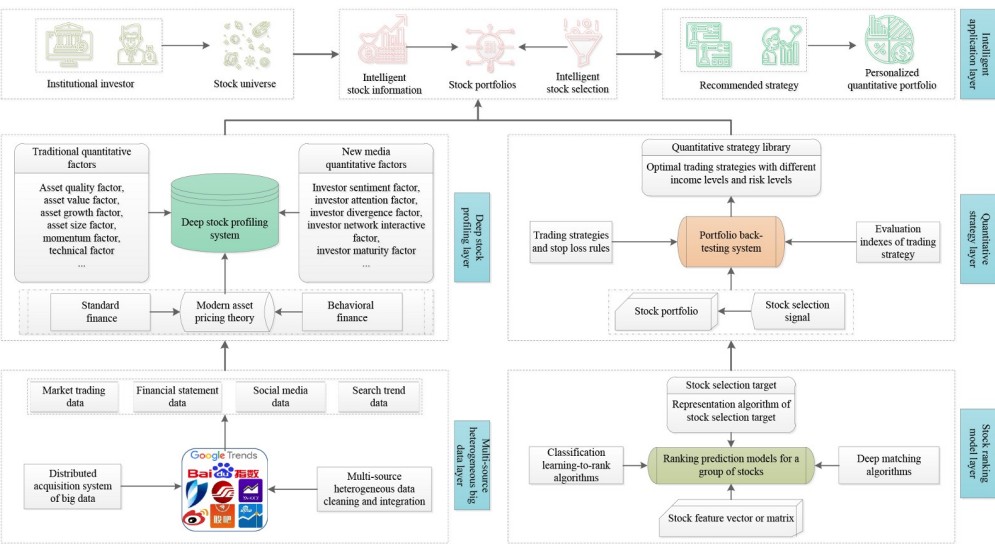

**Fig 16. Intelligent system framework for stock selection.** The framework has a 5-layer architecture, which is developed from left to right and from bottom to top.

## Intelligent system framework for stock selection

The research methods and findings in this paper naturally formed an intelligent system framework for stock selection. As shown in Fig 16, the major modules of this framework were extended from different steps in the research process, ranging from the acquisition of experimental data to the future application of research results.

The main function of the first layer includes collecting financial big data according to data sources and dates, and cleaning and filtering data with complex structures. The second layer corresponds to the deep stock profiling method discussed above. The third layer is an algorithm library used for training ranking or matching models, which pools the optimal ranking learning algorithms and deep matching algorithm. In the fourth layer, the return and risk characteristics of different quantitative strategies are generated through backtesting experiments and the evaluation of trading strategies. The final layer can provide investors with multiple intelligent investment applications, including intelligent information services based on deep stock profiling technology and intelligent stock selection services based on portfolio recommendation technology [90, 91].

The application cases of the intelligent stock selection system can be as follows. The fund managers first select a stock universe, in which the stocks meet certain criterions in terms of age of stocks, liquidity of stocks, and stock popularity in social media. Our intelligent stock selection system will automatically collect and process relevant multi-source heterogeneous big data and construct a profile feature for each stock. Then, according to rebalancing date of the portfolio, deep matching models predict the matching results of the stocks. Next the system marks the portfolios with different returns and risk levels based on the backtesting experiments of the trading strategies. Finally, the fund managers can select suitable portfolios on the basis of their revenue anticipation and risk tolerance, devise personalized trading strategies such as stop loss limit, and obtain their quantitative investment strategies.

## Discussion

In this study, we put forward an innovative deep matching algorithm TS-Deep-LtM for stock portfolio selection using deep stock profiles. The stock matching models trained by TS-Deep-

LtM algorithm were used to capture the relative performance between stocks and to acquire stock selection signals. Based on these signals, we constructed corresponding portfolio strategies, which were demonstrably superior to those of classical LtR algorithms. And this superiority was sustainable across different market conditions.

By comparison of the predictive performances of LtR models based on four different feature combinations, the stock profile feature combination demonstrated its feasibility and effectiveness in addressing the stock selection problem, and was shown as the optimal one. Extracting features from multi-source heterogeneous data simultaneously can better increase the predictive performances of models, when compared to extracting those from a single data source, consistent with the results obtained by Deng et al. and Song et al. They suggested that the training data features extracted from historic price or volume data and news or comments data enabled models to achieve the better predictive effect [28, 92]. Nevertheless, such researches only one-sidedly extract features, such as sentiment factor, attention factor, and technical factor, from two kinds of data sources, and their extraction methods lack of systematic theory support. Our extraction method of deep stock profiling can comprehensively extract the factors affecting stock returns, forming the feature basis for deep matching algorithm.

Dividing the training set by year is the very popular model training method in the low-frequency quantitative investments problem [15, 28]. However, in this study, we found that the trading strategies constructed based on the training datasets of rolling window by samples achieved better annualized return and Sharpe ratio than those constructed based on the training datasets of rolling window by year. As we all known, the future return of a stock is tightly bound to its previous one or several times of changes in returns [19]. In our defined prediction problem, if adopting the dividing method by year, the interval between samples from training and testing sets was excessively long so as to weaken the predictive performance of models and the returns of trading strategies. Instead, adopting the dividing method by samples, stock selection models could be fully trained, especially in the case of limited training data.

Furthermore, by comparison of the predictive performances of nine LtR models, we found that RFRanker models were superior to other LtR models overall. This conclusion conflicts with that obtained by Song et al., in which the LtR models, including RankNet and RankList, trained based on neural network algorithms were found to more applicable to the stock selection problem [28]. This discrepancy might be due to differences in methodology, including feature combination, financial market maturity, or others.

During the training process of TS-Deep-LtM models, the changing tendency of training loss and NDCG@10 reflected a low bias and variance inmodel fitting, indicating TS-Deep-LtM models with a good generalization ability. This might be due to several factors, such as adopting a stock profile feature combination, dividing training data by samples, and setting hyperparameters of preventing overfitting. In addition, we demonstrated the performance of TS-Deep-LtM algorithm preceded that of the RFRanker algorithm in both evaluation metric of models and backtesting results of the trading strategies. The annualized returns of trading strategies constructed by TS-Deep-LtM models were higher than those obtained by the latter (41% vs. 39%), and were higher than those of the top five funds (41% vs. 27%-38%) in China fund industry during the same period according to the statistics from Wind Financial Terminal (https://www.wind.com). The superiority of TS-Deep-LtM algorithm was fully demonstrated. The reasons underlying this superiority might include, but are not limited to, the following: 1) the algorithm tuning and pretraining for a merit-based selection mechanism in TS-Deep-LtM model training, to enhance its adaptability to Non-IID data and the robustness of model's prediction, 2) the matrix data containing more reconstructed signals as the input of TS-Deep-LtM algorithm to alleviate the influence of low SNR data on the training model, and

3) the MV-LSTM algorithm integrated in TS-Deep-LtM algorithm to make it good at handling time series data and capturing signals from financial data.

Based on the experimental procedures and findings, we further proposed an end-to-end design framework of intelligent stock selection system, in which the functional modules correspond to the application practice of quantitative investment. The experimental procedures of stock selection are often scattered and complex; without such a design framework it is difficult to apply the experimental findings in guiding application practice [15, 17, 18, 28]. Although Rasekhschaffe et al. and Fu et al. attempted to propose the methodological frameworks for model constructions [93, 94], yet these frameworks lacked model application modules. Our intelligent stock selection system can potentially act as a robo-advisor, intelligently provide fund managers with quantitative trading strategies satisfying their revenue anticipation and risk tolerance, and produce considerable economic benefit.

The authors admit that the limitations in this study are: 1) using single financial market data, and 2) relative time-consuming matching model training and slightly less cost-effective computational efficiency. For future research, we will continue to explore several possible directions. The same approach can be tried in different stock markets, such as the U.S. stock market, to further extend the availability of the model. We can further enrich the stock profiles. Some new additional factors (i.e., financial hotspot event-related factors) can be added to expand the feature combinations of the training data to optimize the ranking or matching models. Alternatively, if computing power permits, GPU-based and heterogeneous computing-based deep learning methods can be given more attentions and attempts in addressing the similar problems.

## Conclusion

The stock selection task can be transformed into a matching problem between a group of stocks and a stock selection target. The deep matching algorithms can introduce into addressing such problems. Deep stock profile method can combine theoretical and empirical findings from behavioral finance and systematically extract the optimal feature combination from multi-source heterogeneous data, forming the data basis for stock selection models. The stock selection models constructed based on RFRanker algorithm demonstrate superiority in addressing stock selection problem when compared with other learning-to-rank algorithms. The method of dividing training data by samples is more adaptive to the stock selection problem on the limited training data, enabling multiple models to be fully trained. Deep matching algorithm (TS-Deep-LtM) is obtained by setting statistical indicators to filter and integrate three deep text matching algorithms (DRMM, Conv-KNRM, and MV-LSTM). The stock selection models constructed based on TS-Deep-LtM algorithm show the robustness and can efficiently alleviate the challenges posed by the characteristics of a low SNR, the time-series format, and Non-IID financial data. The experimental procedures of stock selection are often scattered and complex; the framework of intelligent stock selection system can provide a relative standard application process for the studies of stock selection problems. More endeavours, such as enriching financial market data, extending stock profiles, and introducing heterogeneous computing architectures, are still needed in stock portfolio selection.

## Supporting information

**S1 File. Methods description for the Chinese text sentiment classification model in financial field and deep text matching algorithms in this study.**
(PDF)

**S2 File. Predicted rankings for a group of stocks in the stock universe by the RFRanker models from 2013-01-07 to 2017-10-16.**
(CSV)

**S3 File. Predicted rankings for a group of stocks in the stock universe by TS-Deep-LtM models from 2013-01-07 to 2017-10-16.**
(CSV)

**S4 File. Feature combination of profile factors and stock profiling data for each stock in the stock universe from 2010-05-10 to 2017-10-16.**
(CSV)

**S5 File. List of stocks in the stock universe, all weekly trading dates for the entire samples from 2013-01-07 to 2017-10-16, and ranking labels of all stocks on each weekly trading date.**
(CSV)

**S6 File. Position adjustment records of trading strategies based on TS-Deep-LtM models in the backtesting experiment from 2013-01 to 2017-10.**
(CSV)

**S7 File. Optimal parameter combinations after hyper-parameters tuning in TS-Deep-LtM model training on a group of training data (2013-01-07).**
(CSV)

**S8 File. Cumulative returns of trading strategies based on TS-Deep-LtM models in the backtesting experiment from 2013-01 to 2017-10.**
(CSV)

**S9 File. Monthly returns of trading strategies based on TS-Deep-LtM models in the back-testing experiment from 2013-01 to 2017-10.**
(CSV)

**S10 File. Alfas of trading strategies based on TS-Deep-LtM models in the backtesting experiment from 2013-01 to 2017-10.**
(CSV)

**S11 File. Betas of trading strategies based on TS-Deep-LtM models in the backtesting experiment from 2013-01 to 2017-10.**
(CSV)

**S12 File. Sharpe ratios of trading strategies based on TS-Deep-LtM models in the backtesting experiment from 2013-01 to 2017-10.**
(CSV)

**S13 File. Volatilities of trading strategies based on TS-Deep-LtM models in the backtesting experiment from 2013-01 to 2017-10.**
(CSV)

**S14 File. Information ratios of trading strategies based on TS-Deep-LtM models in the backtesting experiment from 2013-01 to 2017-10.**
(CSV)

**S15 File. Max drawdowns of trading strategies based on TS-Deep-LtM models in the back-testing experiment from 2013-01 to 2017-10.**
(CSV)

## Acknowledgments

We would like to thank WebShop for its linguistic assistance with an earlier version of the manuscript.

## Author Contributions

**Conceptualization:** Ganggang Guo, Fang Xu.

**Data curation:** Ganggang Guo, Yulei Rao, Feida Zhu, Fang Xu.

**Formal analysis:** Ganggang Guo, Yulei Rao, Feida Zhu.

**Funding acquisition:** Ganggang Guo, Yulei Rao, Fang Xu.

**Investigation:** Ganggang Guo, Yulei Rao, Feida Zhu.

**Methodology:** Ganggang Guo, Yulei Rao, Feida Zhu, Fang Xu.

**Supervision:** Ganggang Guo, Fang Xu.

**Validation:** Ganggang Guo, Feida Zhu.

**Visualization:** Ganggang Guo, Feida Zhu.

**Writing – original draft:** Ganggang Guo, Fang Xu.

**Writing – review & editing:** Ganggang Guo, Yulei Rao, Feida Zhu, Fang Xu.

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
