## [Decision Letter · Decision Letter 0]

5 Jul 2020

PONE-D-20-15612

Innovative deep matching algorithm for stock portfolio selection using deep stock profiles

PLOS ONE

Dear Dr. GUO,

Thank you for submitting your manuscript to PLOS ONE. After careful consideration, we feel that it has merit but does not fully meet PLOS ONE’s publication criteria as it currently stands. Therefore, we invite you to submit a revised version of the manuscript that addresses the points raised during the review process.

The manuscript should be improved towards research contribution, quantitative methods, as well as concluding remarks.

We look forward to receiving your revised manuscript.

Kind regards,

Stefan Cristian Gherghina, PhD. Habil.

Academic Editor

PLOS ONE

Journal Requirements:

Reviewers' comments:

Reviewer's Responses to Questions

**Comments to the Author**

1. Is the manuscript technically sound, and do the data support the conclusions?

Reviewer #1: Yes

Reviewer #2: Partly

2. Has the statistical analysis been performed appropriately and rigorously? 

Reviewer #1: Yes

Reviewer #2: N/A

3. Have the authors made all data underlying the findings in their manuscript fully available?

Reviewer #1: No

Reviewer #2: Yes

4. Is the manuscript presented in an intelligible fashion and written in standard English?

Reviewer #1: Yes

Reviewer #2: Yes

5. Review Comments to the Author

Reviewer #1: Referee Report

Journal: PLOS ONE

Title: Innovative deep matching algorithm for stock portfolio selection using deep stock profiles

Manuscript Number: PONE-D-20-15612

Date: 2020/06/18

Summary

This paper applies the innovative deep matching algorithm to select stock portfolio and test the portfolio strategies. Basically, the topic is interesting for readers interested in stock markets. It is commendable that the authors conduct a vast array of tasks to skillfully introduce their innovative deep matching algorithm, but I still have some reservation about the results of this study. Specific comments are outlined below and I hope my comments can be useful for the authors to enhance this paper.

Major Comments

1. The authors need to be clear on their contribution. I would expect to see a more comprehensive literature review that help readers understand the role of this article in the existing studies. For example, it is necessary to mention clearly what observations are the same or different compared with previous studies using similar approaches. When the observations are different, it is necessary to provide possible explanations for the difference. This could further enhance the contributions of this paper.

2. In methodology, professional technique for stock portfolio selection goes beyond traditional methods so as to be difficult for most readers in the financial field. I would like to recommend the authors to shorten the experimental procedure and lengthen the discussion on the results instead.

3. For empirical results, I suggest the authors to provide economic and financial implications. Providing results are not enough for stock portfolio selection. Readers would like to learn how such results can be applied to their trading strategies. This is an important issue that authors should improve. Please indicate implications and how the general results help stock traders to form investment strategies or conduct market arbitrage. It would make this paper a lot more interesting.

4. The conclusions repeat the operational approach. It would be better for the authors to provide some of concrete takeaways from this paper.

Minor Comments

1. Several figures are blurry.

2. What is the information ratio?

…

Reviewer #2: The document uses a mixture of different tools for natural language processing and information retrieval that are in standard use. Firstly, the document lacks the foundational literature of portfolio theory, as the title suggests that it addresses this problem, the authors suggest that it is based on the factor models developed by FAMA-French but this is not clear, the study bases its motivation on the challenges presented by financial time series regarding a low signal-to-noise ratio which are not well explained, in addition to not explaining from the concept and why it is a problem in finance, it does not explain how its proposed model attacks this problem, since in engineering it is not a minor problem and is addressed in fields such as signal processing in finance is also a relevant problem and deserves to be better explained and do not leave it to the reader. the acronym TS-Deep-LtM, is not clearly explained, Deep given the use of Deep Neural Networks, LtM for Learning to Rank, and TS for Time Series? that is not clear. Treating this topic of the document, there is no evidence on bias and variance assessing to see that the neural networks are not presenting overfitting, since in the financial time series the power of generalization is a key point for the success of a step to production, neither is it presents evidence on whether it was necessary to regularize the model or fine-tuning of the hyperparameters of the neural networks, there is no further evidence on the criteria for performing "feature selection" according to the author, since it is not performed under any objective criteria, the feature selection It is a process that can be done manually or automatically depending on a clear objective. The proposal is novel since it uses NLP methods to address the process of stock selection or stock picking, on this, it must be said that the title does not refer to the content because it is not addressed as a portfolio problem given that the found weightings by the proposed model are not presented and the selection of assets that it makes throughout the analysis sliding window from 2010 to 2017 also, is not presented. About the Data preprocessing could be better, or the paper could document reasons for these particular preprocessing choices. There are several writing sequences that have problems, for example with the use of subjects, plurals and singular in English.

6. PLOS authors have the option to publish the peer review history of their article (what does this mean?). If published, this will include your full peer review and any attached files.

Reviewer #1: No

Reviewer #2: **Yes: **Diego Ismael Leon-Nieto

---

## [Author Response · Author response to Decision Letter 0]

17 Aug 2020

Reviewer #1

Response: Thanks for the professional comments and suggestions. We have revised the manuscript and the changes made were highlighted in a marked-up copy of our manuscript in this way: added text is blue and set in sans-serif, and a red footnote is created for each discarded piece of text. 

1. We have added the discussion section in Page 15 line 584 to Page 16 line 662. A comprehensive literature review is presented to help readers understand the role of this article in the existing studies and enhance the contributions of this paper.

2. We have shortened the experimental procedure, including the construction process of a text sentiment classification model in Page 5 line 191 to Page 6 line 220 and the introduction of deep text matching methods in Page 7 line 245 to Page 8 line 307. We added a discussion section in Page 15 line 584 to Page 16 line 662.

3. In the result section, we added a application scheme of the experimental results in the quantitative investment (Page 15 line 573 to 583). In the intelligent system framework for stock selection, we modified the application module in Fig 16 (Page 14 line 569 to Page 15 line 572). In addition, in the methodology section (Page 8 line 308 to Page 9 line 317), we have revised the part of manuscript to present how to build a portfolio based on deep matching models. In the experiment design section (Page 11 line 406 to 415), we have revised the part of manuscript to present how to design a trading strategy based on a portfolio and to present the weight distribution of stocks and the selection of assets in a trading strategy. They can help readers to learn how such results can be applied to their trading strategies.

4. We have reorganized the conclusion section in Page 16 line 663 to Page 17 line 672.

Minor Comments

1. We have improved the resolution of figures while meeting the requirements of the journal. Please download the original Figures for viewing. 

2. We have added the explanations for the information ratio and other evaluation metrics in Page 12 line 434 to 470.

Reviewer #2

Response: Thanks for the professional comments and suggestions. We have revised the manuscript and the changes made were highlighted in a marked-up copy of our manuscript in this way: added text is blue and set in sans-serif, and a red footnote is created for each discarded piece of text.

1. Firstly, the document lacks the foundational literature of portfolio theory, as the title suggests that it addresses this problem,

Response: We have added the foundational literature of portfolio theory in the introduction section (Page 1 line 1 to Page 2 line 20), which reviews the development of portfolio theory and its practical applications and helps to understand the relationship between the problem posed in this paper and the technical approach we adopted.

2. the authors suggest that it is based on the factor models developed by FAMA-French but this is not clear,

Response: We have revised the part of the manuscript about the factor models in Page 3 line 71 to 77, which clearly states the FAMA-French factor model as a basis of this study.

3. the study bases its motivation on the challenges presented by financial time series regarding a low signal-to-noise ratio which are not well explained, in addition to not explaining from the concept and why it is a problem in finance, it does not explain how its proposed model attacks this problem, since in engineering it is not a minor problem and is addressed in fields such as signal processing in finance is also a relevant problem and deserves to be better explained and do not leave it to the reader.

Response: We have revised the part of the manuscript to explain a low signal-to-noise ratio. The concept of signal-to-noise ratio (Page 2 line 26 to 27), why it is a problem in finance (Page 2 line 27 to 31), and how our proposed model attacks this problem are presented (Page 2 line 56 to Page 3 line 65).

4. the acronym TS-Deep-LtM, is not clearly explained, Deep given the use of Deep Neural Networks, LtM for Learning to Rank, and TS for Time Series? that is not clear. 

Response: We have revised the part of the manuscript about acronym TS-Deep-LtM in Page 4 line 118 to 119, Deep given the use of deep neural networks, LtM for learning-to-match, and TS for Time Series.

5. Treating this topic of the document, there is no evidence on bias and variance assessing to see that the neural networks are not presenting overfitting, since in the financial time series the power of generalization is a key point for the success of a step to production,

Response: In the result section (Page 14 line 535 to 540), we added Figure 14 to evaluate the generalization ability of the deep matching models. The models are not presenting overfitting. We added reasons for the good generalization ability of the models in the discussion section (Page 16 line 622 to 626).

6. neither is it presents evidence on whether it was necessary to regularize the model or fine-tuning of the hyperparameters of the neural networks,

Response: It is necessary to fine-tuning of the hyper-parameters of the neural networks. In the methodology section (Page 8 line 298 to 307), we added the part of the manuscript about the hyper-parameters used in the model tuning phase and corresponding optional parameter values. In the result section (Page 14 line 535 to 537), we added a appendix file (S7 File) to present the optimal parameter combinations generated by the model tuning module during TS-Deep-LtM model training.

7. there is no further evidence on the criteria for performing "feature selection" according to the author, since it is not performed under any objective criteria, the feature selection It is a process that can be done manually or automatically depending on a clear objective.

Response: We have added the description about the criteria for performing "feature selection" in Page 9 line 348 to 351. The "feature selection" criteria for traditional factors is described in Page 9 line 353 to Page 10 line 361. The "feature selection" criteria for social media factors is described in Page 10 line 363 to 374.

8. The proposal is novel since it uses NLP methods to address the process of stock selection or stock picking, on this, it must be said that the title does not refer to the content because it is not addressed as a portfolio problem given that the found weightings by the proposed model are not presented and the selection of assets that it makes throughout the analysis sliding window from 2010 to 2017 also, is not presented.

Response: In the methodology section (Page 8 line 308 to Page 9 line 317), we have revised the part of manuscript to present how to build a portfolio based on deep matching models. In the experiment design section (Page 11 line 406 to 415), we have revised the part of manuscript to present how to design a trading strategy based on a portfolio and to present the weight distribution of stocks and the selection of assets in a trading strategy.

9. About the Data preprocessing could be better, or the paper could document reasons for these particular preprocessing choices.

Response: We have added the part of manuscript (Page 10 line 391 to 393) and Figure 7 to present our specific data preprocessing process in this study.

10. There are several writing sequences that have problems, for example with the use of subjects, plurals and singular in English.

Response: For some writing sequences that have problems, we have made corresponding modifications and they are highlighted in a marked-up copy of our manuscript in this way: added text is blue and set in sans-serif, and a red footnote is created for each discarded piece of text.

---

## [Decision Letter · Decision Letter 1]

29 Sep 2020

PONE-D-20-15612R1

Innovative deep matching algorithm for stock portfolio selection using deep stock profiles

PLOS ONE

Dear Dr. GUO,

Thank you for submitting your manuscript to PLOS ONE. After careful consideration, we feel that it has merit but does not fully meet PLOS ONE’s publication criteria as it currently stands. Therefore, we invite you to submit a revised version of the manuscript that addresses the points raised during the review process. Further revisions towards paper structure and several explanations are required.

We look forward to receiving your revised manuscript.

Kind regards,

Stefan Cristian Gherghina, PhD. Habil.

Academic Editor

PLOS ONE

Reviewers' comments:

Reviewer's Responses to Questions

**Comments to the Author**

1. If the authors have adequately addressed your comments raised in a previous round of review and you feel that this manuscript is now acceptable for publication, you may indicate that here to bypass the “Comments to the Author” section, enter your conflict of interest statement in the “Confidential to Editor” section, and submit your "Accept" recommendation.

Reviewer #1: (No Response)

Reviewer #3: All comments have been addressed

2. Is the manuscript technically sound, and do the data support the conclusions?

Reviewer #1: (No Response)

Reviewer #3: Yes

3. Has the statistical analysis been performed appropriately and rigorously? 

Reviewer #1: (No Response)

Reviewer #3: N/A

4. Have the authors made all data underlying the findings in their manuscript fully available?

Reviewer #1: (No Response)

Reviewer #3: Yes

5. Is the manuscript presented in an intelligible fashion and written in standard English?

Reviewer #1: (No Response)

Reviewer #3: Yes

6. Review Comments to the Author

Reviewer #1: (No Response)

Reviewer #3: Based on my review; The authors have provided the requested information by the reviewers and I would like to suggest this article is accepted. But, the authors needs to improve algorithm representations for the Algorithm 1 and 2. It will make the readers easy to know the algorithm flow.

7. PLOS authors have the option to publish the peer review history of their article (what does this mean?). If published, this will include your full peer review and any attached files.

Reviewer #1: No

Reviewer #3: No

---

## [Author Response · Author response to Decision Letter 1]

5 Oct 2020

Response: Thanks for the professional comments and suggestions. We have revised the manuscript and the changes made were highlighted in a marked-up copy of our manuscript in this way: added text is blue and set in sans-serif, and a red footnote is created for each discarded piece of text. 

1. We have improved algorithm representations for the Algorithm 1 in Page 6 and the Algorithm 2 in Page 8.

2. We have shortened the abstract section and removed some details in Page 1. Appropriate extensions have been made and economic intuitions for the main results were summarized in the conclusion section in Page 17 line 680 to 699. 

3. We have explained the roles of evaluation metrics of trading strategies in stock portfolio selection and how these metrics affect stock portfolio selection in Page 13 line 471 to 486.

---

## [Decision Letter · Decision Letter 2]

19 Oct 2020

Innovative deep matching algorithm for stock portfolio selection using deep stock profiles

PONE-D-20-15612R2

Dear Dr. GUO,

We’re pleased to inform you that your manuscript has been judged scientifically suitable for publication and will be formally accepted for publication once it meets all outstanding technical requirements.

Kind regards,

Stefan Cristian Gherghina, PhD. Habil.

Academic Editor

PLOS ONE

Additional Editor Comments (optional):

Reviewers' comments:

Reviewer's Responses to Questions

**Comments to the Author**

1. If the authors have adequately addressed your comments raised in a previous round of review and you feel that this manuscript is now acceptable for publication, you may indicate that here to bypass the “Comments to the Author” section, enter your conflict of interest statement in the “Confidential to Editor” section, and submit your "Accept" recommendation.

Reviewer #1: All comments have been addressed

Reviewer #3: All comments have been addressed

2. Is the manuscript technically sound, and do the data support the conclusions?

Reviewer #1: Partly

Reviewer #3: Yes

3. Has the statistical analysis been performed appropriately and rigorously? 

Reviewer #1: (No Response)

Reviewer #3: N/A

4. Have the authors made all data underlying the findings in their manuscript fully available?

Reviewer #1: No

Reviewer #3: Yes

5. Is the manuscript presented in an intelligible fashion and written in standard English?

Reviewer #1: Yes

Reviewer #3: Yes

6. Review Comments to the Author

Reviewer #1: I agree that the authors have done a good job in the previous round and the quality of this paper has been considerably improved. There are no other comments in this version.

Reviewer #3: All comments have been addressed. The authors have provided the information required by the reviewer.

7. PLOS authors have the option to publish the peer review history of their article (what does this mean?). If published, this will include your full peer review and any attached files.

Reviewer #1: No

Reviewer #3: No

---

## [Editor Report · Acceptance letter]

23 Oct 2020

PONE-D-20-15612R2 

Innovative deep matching algorithm for stock portfolio selection using deep stock profiles 

Dear Dr. GUO:

I'm pleased to inform you that your manuscript has been deemed suitable for publication in PLOS ONE. Congratulations! Your manuscript is now with our production department. 

Kind regards, 

on behalf of

Dr. Stefan Cristian Gherghina 

Academic Editor

PLOS ONE